# Causal LLM Routing: End-to-End Regret Minimization from Observational Data

**Asterios Tsiourvas**
MIT

**Wei Sun**
IBM Research

**Georgia Perakis**
MIT

## Abstract

LLM routing aims to select the most appropriate model for each query, balancing competing performance metrics such as accuracy and cost across a pool of language models. Prior approaches typically adopt a *decoupled* strategy, where metrics are first predicted and the model is then selected based on these estimates. This setup is prone to compounding errors and often relies on *full-feedback* data, where each query is evaluated by all candidate models, which is costly to obtain and maintain in practice. In contrast, we learn from *observational data*, which records only the outcome of the model actually deployed. We propose a *causal end-to-end* framework that learns routing policies by minimizing decision-making regret from observational data. To enable efficient optimization, we introduce two theoretically grounded surrogate objectives: a classification-based upper bound, and a softmax-weighted regret approximation shown to recover the optimal policy at convergence. We further extend our framework to handle heterogeneous cost preferences via an interval-conditioned architecture. Experiments on public benchmarks show that our method outperforms existing baselines, achieving state-of-the-art performance across different embedding models.

## 1 Introduction

LLM routing is an emerging research area focused on optimizing model selection for each input query, balancing performance and cost across a pool of available LLMs. Because LLM performance varies significantly by task and input [Hu et al., 2024], as well as computational cost [Ong et al., 2024], dynamic routing strategies have been proposed to select the most suitable model per query [Shnitzer et al., 2023]. This challenge becomes even more critical in agentic applications, where multiple LLM calls may be made within a single workflow, making efficient model selection essential for user experience and resource allocation. As LLM deployment scales, routing also contributes to environmental sustainability by reducing unnecessary computation [Singh et al., 2025].

Routing methods can be broadly classified based on whether they invoke one or multiple models per query. *Multi-model* approaches include *non-predictive routing*, which cascades models sequentially from light to heavy [Wang et al., 2023, Chen et al., 2023], and *predictive ensembles*, which learn to combine outputs or scores from multiple LLMs [Shekhar et al., 2024, Huang et al., 2025]. While these methods can improve accuracy and robustness, they incur significant computational cost and latency due to repeated inference. By contrast, *predictive routing* methods, including our approach, select a single LLM for each query by training a router that maps input queries to the most appropriate model [Shnitzer et al., 2023, Ong et al., 2024, Somerstep et al., 2025]. This framework offers a more scalable and cost-efficient solution, particularly in settings where minimizing latency and compute resources is critical.

A common formulation in predictive routing is to maximize a utility function of the form $y_x(t) = a_x(t) - \lambda \cdot c_x(t)$, where $a_x(t)$ and $c_x(t)$ denote the quality (e.g., accuracy) and cost for model $t$ on a given query $x$, and $\lambda \geq 0$ captures user cost sensitivity, or willingness-to-pay. Existing methods

39th Conference on Neural Information Processing Systems (NeurIPS 2025).

typically adopt a *decoupled* approach: separate predictors are trained for each metric, and routing is performed by selecting the model with the highest estimated utility. However, decision quality is highly sensitive to these predictors, and errors can compound, especially when incorporating additional metrics (e.g., latency, faithfulness, alignment), increasing complexity and uncertainty.

Another fundamental limitation in the existing literature on predictive routing is its reliance on *full-feedback* datasets. Prior work [Ong et al., 2024] [Somerstep et al., 2025] assumes access to data where each query has been evaluated by all available LLMs. This assumption is impractical: (i) the computational and monetary cost of exhaustively querying all models is prohibitive; and (ii) the rapid pace of LLM development makes it challenging to maintain comprehensive, up-to-date evaluation datasets. In contrast, *observational data*, where each query is evaluated by only one model, is readily available from real-world LLM deployments, making it far more scalable than full feedback datasets. However, it introduces *treatment bias* from historical routing policies, which can lead to suboptimal decisions if not properly addressed [Swaminathan and Joachims, 2015, Künzel et al., 2019].

To the best of our knowledge, this is the first work to (i) learn LLM routing from observational data and (ii) introduce an integrated learning framework for routing. Our main contributions are:

- We propose a *causal end-to-end* framework that learns routing policies by directly minimizing decision-making regret using observational data. Unlike the predominant decoupled paradigm, where various performance metrics (e.g., accuracy, cost) are first predicted and then used to inform routing decisions, our method integrates prediction and prescription into a unified objective. By optimizing for regret directly, the framework is explicitly aligned with the final routing decision quality. Furthermore, it is designed to scale efficiently and leverage readily available observational data, while accounting for treatment bias without requiring costly full-feedback datasets.

- As the original regret minimization objective is not directly differentiable, we derive two *surrogate objectives* to enable end-to-end policy learning. The first is a classification-based upper bound that reframes regret minimization as a multiclass prediction problem under mild Lipschitz assumptions, allowing efficient training with standard methods. The second is a softmax-weighted regret surrogate that smoothly approximates regret using a softmax distribution and provably recovers optimal decisions at convergence.

- We extend our framework to support *heterogeneous cost preferences* by introducing a unified model that conditions on both the query and the user's cost sensitivity. Leveraging the affine structure of the utility function, we design an efficient interpolation scheme using only two endpoint models per interval. We theoretically show that the optimal treatment is piecewise constant in the cost parameter and that our architecture can exactly represent the optimal policy, enabling flexible and scalable routing across diverse preferences.

- We conduct *comprehensive experiments* on two public benchmarks, demonstrating that our regret-minimizing and heterogeneous cost-aware approaches consistently outperform existing baselines. Our methods achieve state-of-the-art performance across both BERT-based and LLaMA-based embeddings, highlighting their robustness and practical effectiveness.

## 2 Methodology

### 2.1 Problem Formulation

We consider a dataset of $n$ observational samples, denoted by $\mathcal{D} = \{(x_i, t_i, a_i, c_i)\}_{i=1}^n$, where each sample is independently drawn from the joint distribution $p(x, t, a, c)$. Here, $x_i \in \mathcal{X} \subset \mathbb{R}^d$ denotes a feature vector, typically an embedding, that characterizes the query; $t_i \in [\mathcal{T}] := \{0, 1, \ldots, T-1\}$ specifies the LLM assigned to the query; $a_i \in \mathbb{R}_{\geq 0}$ denotes a numeric quality score of the LLM's response, such as accuracy, or a preference rating; and $c_i \in \mathbb{R}_{\geq 0}$ represents the cost incurred by model $t_i$ when processing query $x_i$.

Given a query $x \in \mathcal{X}$, the objective of an LLM router is to select a model $t \in \mathcal{T}$ that maximizes the cost-aware performance, or utility, defined as $y_x(t) := a_x(t) - \lambda \cdot c_x(t)$. Here, $\lambda \geq 0$ is a user-specified parameter, modeling the trade-off between accuracy and cost. Higher values of $y_x(t)$, corresponding to greater performance, are preferred.

## 2.2 End-to-End Regret Minimization

Our goal is to route each prompt $x$ to the LLM $t$ such that the decision leads to the highest possible utility $y_x(t)$. To this end, we aim to learn an end-to-end policy $f : \mathcal{X} \to \mathcal{T}$ that minimizes the decision-making regret [Fernández-Loría and Provost, 2022, Zou et al., 2022]. More formally,

$$f^* := \arg\min_f Regret(f)$$

where regret is defined as

$$Regret(f) := \mathbb{E}_X[Y_X(t_X^*) - Y_X(f(X))] = \mathbb{E}_X[Y_X(t_X^*)] - \mathbb{E}_X[Y_X(f(X))], \tag{1}$$

with $t_X^* := \arg\max_{t \in \mathcal{T}} Y_X(t)$.

We want to point out that unlike full-feedback datasets used in prior routing work, which record outcomes for all models $t \in \mathcal{T}$, observational datasets contain only *partial feedback*, logging the outcome of a single model $t_i$ selected by historical policies. As a result, counterfactual outcomes for unobserved LLMs are missing, making it necessary to estimate them while correcting for treatment bias. We address this challenge of estimating $\hat{Y}_X(\cdot)$ using causal inference techniques, as detailed in Section 2.3.

With an accurate approximation $\hat{Y}_X(\cdot)$, the empirical regret can be approximated as

$$Regret(f) \approx \frac{1}{n} \sum_{i=1}^n \left( \hat{Y}_{x_i}(t_i^*) - \hat{Y}_{x_i}(f(x_i)) \right), \tag{2}$$

where $t_i^* := \arg\max_{t \in \mathcal{T}} \hat{Y}_{x_i}(t)$ is the estimated optimal decision for query $x_i$.

A key challenge with the objective in Equation (2) is its dependence on the discrete routing decision $f(x_i)$, which makes the regret non-differentiable. To address this, we introduce two surrogate loss functions that serve as differentiable approximations.

**Surrogate Loss 1: Classification-Based Upper Bound**   Our first approach is to derive a tractable upper bound on the regret and directly minimize it. To do so, we define the following notion of Lipschitz continuity for utility functions over the probability simplex.

**Definition 1.** *Let $\hat{Y}_x : \mathcal{T} \to \mathbb{R}$ be an estimated utility function assigning a scalar utility to each model $t \in \mathcal{T}$ for a given input $x$. We extend $\hat{Y}_x$ to the probability simplex $\Delta^{|\mathcal{T}|}$ by defining*

$$\hat{Y}_x(p) := \sum_{t \in \mathcal{T}} p(t)\hat{Y}_x(t), \tag{3}$$

*for any $p \in \Delta^{|\mathcal{T}|}$. We say that $\hat{Y}_x$ is L-Lipschitz over the simplex with respect to the $\ell_1$ norm if for all $p, q \in \Delta^{|\mathcal{T}|}$,*

$$|\hat{Y}_x(p) - \hat{Y}_x(q)| \le L \cdot \|p - q\|_1. \tag{4}$$

*Here, the constant L can be taken as $L := \max_{t \in \mathcal{T}} |\hat{Y}_x(t)|$.*

This condition ensures that small changes in the model distribution lead to bounded changes in expected utility. Since $\mathcal{T}$ is a finite set and $\hat{Y}_x(\cdot)$ is typically learned via bounded smooth function approximators (e.g., neural networks), it is natural to expect bounded variation in utility values across nearby treatments. Note that when $p$ is a one-hot vector $e_{t^*}$ and $q = f(x)$ is a probabilistic policy, this setting corresponds to our model selection problem, where we seek to minimize the regret between the optimal choice and a stochastic routing decision.

*Proposition* 1. Suppose the estimated utility function $\hat{Y}_x : \mathcal{T} \to \mathbb{R}$ is $L$-Lipschitz continuous over the probability simplex with respect to the $\ell_1$ norm, as in Definition 1. Then, for a policy $f : \mathcal{X} \to \Delta^{|\mathcal{T}|}$ that outputs a distribution $f(x)$ over $\mathcal{T}$, the regret can be upper bounded by:

$$\text{Regret}(f) \le L \cdot \frac{1}{n} \sum_{i=1}^n \sqrt{2 \cdot \text{CE}(t_i^*, f(x_i))}, \tag{5}$$

where $t_i^* := \arg\max_{t \in \mathcal{T}} \hat{Y}_{x_i}(t)$ is the optimal treatment for input $x_i$, and $\text{CE}(t_i^*, f(x_i)) := -\log f(x_i)_{t_i^*}$ denotes the cross-entropy loss.

This motivates a classification-based surrogate objective: rather than modeling the full utility surface, we directly learn a policy $f : \mathcal{X} \to \mathcal{T}$ by solving a supervised learning problem, where the target label for each input $x_i$ is the estimated optimal decision $t_i^*$. Optimizing a classification loss $d(t_i^*, f(x_i))$, serves as a tractable surrogate for minimizing the regret in Equation (2), as it upper bounds the regret under mild assumptions. This formulation reduces policy learning to a multiclass classification task, enabling efficient training using standard techniques.

### 2.2.1 Surrogate Loss 2: Softmax-Weighted Regret

The second proxy directly minimizes the regret using a differentiable softmax approximation. Specifically, we model the policy function $f$ as a neural network with $|\mathcal{T}|$ outputs passed through a softmax layer with temperature parameter $\tau > 0$, which makes the regret surrogate in Equation (2) differentiable. The first term of the regret, $\mathbb{E}_X[Y_X(t^*)]$, is approximated as:

$$\mathbb{E}_X[Y_X(t^*)] \approx \frac{1}{n} \sum_{i=1}^{n} \hat{Y}_{x_i}(t_i^*), \quad \text{where } t_i^* := \arg\max_{t \in \mathcal{T}} \hat{Y}_{x_i}(t). \tag{6}$$

The second term, $\mathbb{E}_X[Y_X(f(X))]$, is estimated by treating the softmax output as a distribution over treatments:

$$\mathbb{E}_X[Y_X(f(X))] \approx \frac{1}{n} \sum_{i=1}^{n} \sum_{t=1}^{|\mathcal{T}|} \hat{Y}_{x_i}(t) \cdot \text{softmax}(f(x_i))_t. \tag{7}$$

Combining the two, we minimize the following differentiable surrogate objective:

$$\min_f \frac{1}{n} \sum_{i=1}^{n} \left( \hat{Y}_{x_i}(t_i^*) - \sum_{t=1}^{|\mathcal{T}|} \hat{Y}_{x_i}(t) \cdot \text{softmax}(f(x_i))_t \right). \tag{8}$$

After training, the learned policy prescribes for each $x \in \mathcal{X}$ the treatment $\hat{t} = \arg\max_{t \in \mathcal{T}} f(x)_t$. We now show that this objective recovers pointwise optimal treatment assignment, thus providing a consistent and differentiable approximation to the original regret minimization objective.

*Proposition* 2. Let $f : \mathcal{X} \to \mathbb{R}^{|\mathcal{T}|}$ be a neural network whose output is passed through a softmax layer with fixed temperature $\tau > 0$, and define $t_i^* := \arg\max_{t \in \mathcal{T}} \hat{Y}_{x_i}(t)$. Then, optimizing the softmax-weighted surrogate regret objective via gradient descent

$$\min_f \frac{1}{n} \sum_{i=1}^{n} \left( \hat{Y}_{x_i}(t_i^*) - \sum_{t=1}^{|\mathcal{T}|} \hat{Y}_{x_i}(t) \cdot \text{softmax}(f(x_i))_t \right) \tag{9}$$

leads the model $f$ to concentrate all probability mass on the optimal treatment $t_i^*$. That is, at convergence,

$$\text{softmax}(f(x_i))_t \to \begin{cases} 1 & \text{if } t = t_i^*, \\ 0 & \text{otherwise.} \end{cases} \tag{10}$$

### 2.3 Estimating Counterfactual Utility via Causal Inference

In the previous sections, we assumed that we have access to $\hat{Y}_X(\cdot)$. Given the observational nature of the data, the potential utility function $Y_x(\cdot)$ is not directly observable. We follow the potential outcomes framework Rosenbaum and Rubin [1983], Rubin [1984] and assume the existence of a potential utility function $Y_x(t)$. We adopt the following assumptions, which are standard in the causal inference literature.

*Assumption* 1 (Stable Unit Treatment Value). The potential outcome of one sample is independent of the treatment assignments on the other samples.

*Assumption* 2 (Ignorability). The assigned treatments and potential outcomes are independent conditional on observed covariates, i.e. $t \perp \{Y_x(t')|t' \in \mathcal{T}\}|x$ [Hirano and Imbens, 2004].

*Assumption* 3 (Support). For $x \in \mathcal{X}$ such that $p(x) > 0$, we have that $p(t|x) > 0$ for each $t \in \mathcal{T}$.

In the causal inference literature, counterfactual outcomes can be estimated using various methods. Such examples include the "meta-learner" [Künzel et al., 2019], or the Inverse Propensity Weighting

(IPW) estimator [Horvitz and Thompson, 1952]. In this work, we utilize the doubly robust estimator introduced by Dudík et al. [2011], defined as:

$$\hat{Y}_x(t) := \frac{(y - \hat{r}_t(x))\mathbb{1}[\pi(x) = t]}{\hat{p}(t|x)} + \hat{r}_t(x), \quad \forall t \in \mathcal{T}, \tag{11}$$

where $\hat{r}_t : \mathcal{X} \to \mathcal{Y}$ denotes the direct outcome regression model for treatment $t$, $\hat{p}(t|x)$ is the estimated propensity score, and $\pi : \mathcal{X} \to \mathcal{T}$ is the logging policy observed in the dataset $\mathcal{D}$.

The doubly robust estimator combines an outcome model $\hat{r}_t(x)$ and a propensity model $\hat{p}(t|x)$, yielding consistent estimates if either is correctly specified [Dudík et al., 2011]. It offers a favorable bias-variance trade-off: the propensity model corrects for selection bias, while the outcome model reduces variance by leveraging structure in the data.

*Remark* 1. While we use the doubly robust (DR) estimator in our experiments due to its favorable bias–variance tradeoff and strong practical performance, our framework is estimator-agnostic: any valid counterfactual estimator can be used to compute utility estimates. This includes more advanced approaches that relax or mitigate these assumptions

In the experimental section, we show that ignoring the treatment bias leads to inaccurate counterfactual estimates and causes substantial degradation in routing quality, highlighting the limitations of standard supervised learning approaches that assume full feedback.

## 3    Routing under Heterogeneous Cost Preferences

In the previous section, we introduced a causal end-to-end framework for learning optimal routing policies from observational data, where the objective is to maximize a utility function of the form $y = a - \lambda c$, with a *fixed* $\lambda \geq 0$ representing the trade-off between accuracy and cost.

In practice, however, user preferences vary, i.e., different queries may be associated with different values of $\lambda$. From a system design perspective, training and maintaining a separate router for each possible $\lambda$ is impractical. In this section, we propose a unified approach that supports routing under heterogeneous cost sensitivities. We first present a joint model architecture that conditions on both the query and the cost parameter, and then provide a theoretical analysis to justify the proposed design.

### 3.1    Interval-Conditioned Joint Router

We design a neural network $f : \mathcal{X} \times \mathbb{R}_{\geq 0} \to \mathbb{R}^{|\mathcal{T}|}$ that jointly takes a query $x \in \mathcal{X}$ and a cost sensitivity parameter $\lambda \in \mathbb{R}_{\geq 0}$ as input, outputs a score vector over available LLMs and the routing decision is then made via:

$$\hat{t} = \arg \max_{t \in \mathcal{T}} f(x, \lambda)_t. \tag{12}$$

We assume access to a finite, representative set of cost preferences $\Lambda := \{\lambda_1, \ldots, \lambda_m\} \subset \mathbb{R}_{\geq 0}$. For ease of notation we assume that $\lambda_1 < \lambda_2 < \cdots < \lambda_m$. In practice, these values may correspond to discrete service quality tiers (e.g., basic, standard, premium) that reflect users' varying willingness to trade off cost for performance. For each $\lambda \in \Lambda$, we partition the training data by cost preference and estimate $\lambda$-specific utility $\hat{Y}_{x_i}^{\lambda}(t)$, which forms the basis of our joint interval-conditioned architecture.

**Training Procedure.**

1. For each $\lambda \in \Lambda$, we first train an individual router $f_\lambda : \mathcal{X} \to \mathbb{R}^{|\mathcal{T}|}$ using the methods introduced in Section 2.2.
2. For each interval $[\lambda_j, \lambda_{j+1}]$ $j \in [1, \ldots, m-1]$, we initialize a joint network $f(x, \lambda) : \mathcal{X} \times \mathbb{R}_{\geq 0} \to \mathbb{R}^{|\mathcal{T}|}$ that uses as input both $x, \lambda \in (\lambda_j, \lambda_{j+1})$.
3. The shared model is fine-tuned to minimize regret over the interval:

$$\min_f \frac{1}{2n} \sum_{\lambda \in \{\lambda_j, \lambda_{j+1}\}} \sum_{i=1}^n \text{Regret}\left(f(x_i, \lambda)\right), \tag{13}$$

where regret is computed using the doubly robust estimator as described earlier, under the corresponding $\lambda$-specific utility $\hat{Y}_{x_i}^{\lambda}(t)$.

**Deployment Strategy.** At inference time, given a user-specified cost sensitivity parameter $\lambda \in \mathbb{R}_{\geq 0}$:

- If $\lambda \in \Lambda$, we use the individual model $f_\lambda(x)$.
- If $\lambda \notin \Lambda$, we identify the closest neighbors $\underline{\lambda}, \overline{\lambda} \in \Lambda$ such that $\underline{\lambda} < \lambda < \overline{\lambda}$, and we use the corresponding joint network $f(x, \lambda)$ trained to generalize across the preference in $(\underline{\lambda}, \overline{\lambda})$.

## 3.2 Model Architecture

A key component of the heterogeneous cost preference routing setup introduced in Section 3.1 is the interval-conditioned joint model $f(x, \lambda)$, that for each interval $[\lambda_j, \lambda_{j+1}]$, interpolates between the two corresponding individual models $f_{\lambda_j}$ and $f_{\lambda_{j+1}}$. Specifically, the architecture is designed to exploit the *affine structure* of the utility function with respect to $\lambda$, namely $y = a - \lambda c$. This motivates a lightweight parameterization that uses only the two endpoints of the interval $[\lambda_j, \lambda_{j+1}]$ rather than all $m$ pre-trained models. Concretely, the joint model that we propose is defined as:

$$f(x, \lambda) = \mathtt{Linear}\left([f_{\lambda_j}(x), \ f_{\lambda_{j+1}}(x)] + \ g(\lambda)\right), \tag{14}$$

where $[\cdot, \cdot]$ denotes concatenation, and $g(\lambda) := \mathtt{Activation}(\mathtt{Linear}(\lambda))$ is a learnable representation of the cost sensitivity parameter.

This architecture enables smooth interpolation between $f_{\lambda_j}$ and $f_{\lambda_{j+1}}$ within $[\lambda_j, \lambda_{j+1}]$, allowing the router to adapt to intermediate values of $\lambda$ without requiring an individual model for each one. By conditioning only on the two bounding models, this design achieves computational efficiency and strong generalization across cost preferences. The proposed architecture is illustrated in Figure 1.

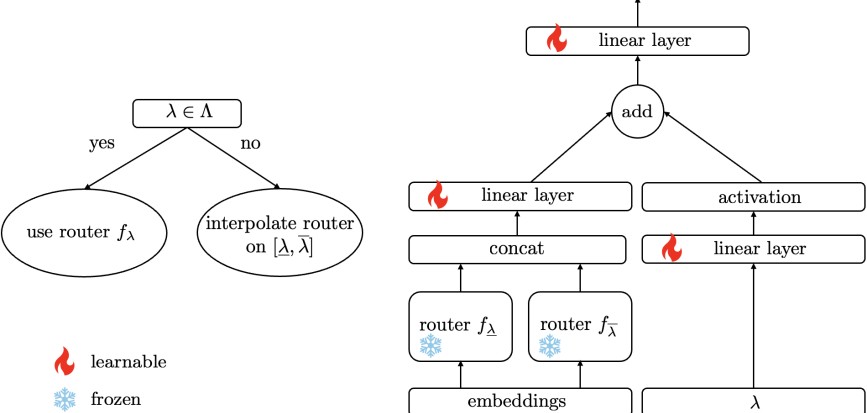

Figure 1: Overview of the proposed interval-conditioned joint router framework. **Left:** Decision logic for handling a given cost sensitivity parameter $\lambda$. **Right:** Joint router architecture.

## 3.3 Theoretical Guarantees

We now present theoretical guarantees that justify the structure and training strategy of joint cost-preference router. These guarantees leverage the affine nature of the utility function, which is linear in the cost parameter $\lambda$ for fixed accuracy $a$ and cost $c$, enabling exact interpolation for specific cost sensitivity across fixed intervals.

*Proposition* 3 (Piecewise Constant Optimal Policy). Fix a query $x \in \mathcal{X}$ and assume the estimated utility is affine in $\lambda$, i.e., $\hat{Y}_x^\lambda(t) = a_x(t) - \lambda \cdot c_x(t)$ for all $t \in \mathcal{T}$. Then the optimal treatment

$$t^*(\lambda) := \arg\max_{t \in \mathcal{T}} \hat{Y}_x^\lambda(t)$$

is piecewise constant in $\lambda$. That is, the cost parameter $\mathbb{R}_{\geq 0}$ can be partitioned into intervals over which the optimal treatment remains fixed.

*Proposition* 4 (Affine Closure of Utility Function). Let $\lambda_j < \lambda_{j+1}$ be two adjacent cost values and let $\lambda \in [\lambda_j, \lambda_{j+1}]$. Suppose the utility function is affine in $\lambda$, i.e., $\hat{Y}_x^\lambda(t) = a_x(t) - \lambda \cdot c_x(t)$. Then

for all $t \in \mathcal{T}$, the utility at $\lambda$ is a convex combination of utilities at the endpoints:

$$\hat{Y}_x^\lambda(t) = \alpha \cdot \hat{Y}_x^{\lambda_j}(t) + (1 - \alpha) \cdot \hat{Y}_x^{\lambda_{j+1}}(t), \quad \text{where } \alpha := \frac{\lambda_{j+1} - \lambda}{\lambda_{j+1} - \lambda_j}.$$

*Corollary* 1 (Sufficiency of Two Models per Interval). Under the affine assumption, the utility $\hat{Y}_x^\lambda(t)$ for any $\lambda \in [\lambda_j, \lambda_{j+1}]$ can be exactly reconstructed using only the endpoints $\hat{Y}_x^{\lambda_j}(t)$ and $\hat{Y}_x^{\lambda_{j+1}}(t)$. Thus, it is sufficient to use only the two corresponding models $f_{\lambda_j}$ and $f_{\lambda_{j+1}}$ for interpolation within the interval.

*Proposition* 5 (Expressivity of Additive Two-Model joint Architecture). Let $\lambda \in [\lambda_j, \lambda_{j+1}]$, and suppose that for each $t \in \mathcal{T}$ the utility satisfies $\hat{Y}_x^\lambda(t) = a_x(t) - \lambda \cdot c_x(t)$. Then the optimal treatment $t^*(\lambda) := \arg\max_t \hat{Y}_x^\lambda(t)$ can be exactly represented by a softmax policy over a function of the form:

$$f(x, \lambda) = \texttt{Linear}\left([f_{\lambda_j}(x), f_{\lambda_{j+1}}(x)] + g(\lambda)\right),$$

where $g(\lambda)$ is any differentiable embedding of $\lambda$, and $f_{\lambda_j}, f_{\lambda_{j+1}}$ are accurate predictors trained at endpoints $\lambda_j$ and $\lambda_{j+1}$.

**Implications for Architecture and Training.** The theoretical results above provide strong justification for both the proposed model architecture and the associated training procedure. Proposition 3 shows that the optimal treatment changes only across a small number of cost sensitivities, supporting our interval-conditioned strategy for routing. Proposition 4 guarantees that the utility for any intermediate cost preference can be exactly recovered through convex interpolation of endpoint models. Finally, Proposition 5 establishes that our joint model is expressive enough to capture the optimal policy within each interval. Our method provides a principled approach for learning a joint routing model that accommodates heterogeneous cost preferences from observational data.

## 4 Experiments

### 4.1 Datasets

We evaluate our methods on two publicly available benchmarks for LLM routing: **RouterBench** [Hu et al., 2024] and **SPROUT** [Somerstep et al., 2025].

**RouterBench** is a standardized benchmark comprising 35,712 prompt–response pairs from 11 language models. The prompts are drawn from eight evaluation suites spanning reasoning, factual recall, dialogue, mathematics, and code generation. Each prompt is annotated with model accuracy and execution cost, enabling supervised training and evaluation of routing policies.

**SPROUT** is a larger and more diverse benchmark focused on cost-aware routing, consisting of 44,241 prompts and responses from 13 state-of-the-art LLMs. The prompts cover six challenging tasks: GPQA [Rein et al., 2024], MuSR [Sprague et al., 2023], MMLU-Pro [Wang et al., 2024], MATH [Hendrycks et al., 2021], OpenHermes [Teknium, 2023], and RAGBench [Friel et al., 2024]. SPROUT includes a predefined split: 80% for training, with the remaining 20% evenly divided between validation and test sets.

### 4.2 Embeddings and Model Architecture

To encode input queries into vector representations $x$, we generate embeddings using two compact, publicly available language models: `BERT-base-uncased` (768 dimensions) and `Llama-3.2-1B` (2048 dimensions). Each input is passed once through the model, and the final hidden states are mean-pooled to obtain a fixed-length embeddings. These models were selected for their efficiency and suitability for real-time routing.

The embeddings are processed by a two-layer fully connected network with GELU activations and 200 hidden units per layer. The model is trained with the Adam optimizer (learning rate $1 \times 10^{-4}$) for up to 10,000 epochs, using early stopping with a patience of 100. A softmax output with temperature $\tau = 100$ is used to control the sharpness of the output probabilities. This architecture is used consistently across all benchmarked methods for fair comparison. For the doubly robust estimator, the same network models the direct outcomes $\hat{r}_t(x)$, while the propensity scores $\hat{p}(t \mid x)$ are estimated

using XGBoost. To reduce variance from extreme inverse propensity weights, we apply clipping at the 5th and 95th percentiles. The only architectural modification is for the interval-based model, where the softmax temperature is increased to $\tau = 1000$ to enable smoother interpolation across $\lambda$. Hyperparameters are summarized in Appendix D.

## 4.3 Methods

We evaluate our proposed routing strategies against a range of baselines from the causal machine learning and LLM routing literature. Since both SPROUT and RouterBench provide full-feedback datasets (i.e., responses from all models), we simulate observational data by sampling a single model per prompt. Specifically, for each prompt, we sample a model $t \in \mathcal{T}$ with probability proportional to its accuracy $\mathbb{P}[t = \tau] = \frac{e^{a_\tau}}{\sum_{\tau' \in \mathcal{T}} e^{a_{\tau'}}}$, where $a_\tau$ is the accuracy of model $\tau$ on that prompt.

As an optimistic oracle, we include a **Full-Feedback** model that learns a model $f : \mathcal{X} \to \mathbb{R}^{|\mathcal{T}|}$ using the complete outcome vector for each query and optimizes a standard multi-class classification loss. We benchmark against a common decoupled routing strategy denoted **Baseline**, which independently estimates model accuracy $a_x(t)$ and cost $c_x(t)$, without accounting for selection bias. This reflects the approach taken in prior predictive routing methods such as CARROT [Somerstep et al., 2025], which has demonstrated superior performance over alternatives like RouteLLM [Ong et al., 2024] and RoRF [Jain et al., 2023]. To adjust for treatment assignment bias, we consider a **Regress-and-Compare (R&C)** method, which fits outcome models $\hat{Y}_x(t)$ for each treatment $t$, and selects the action $\hat{t} = \arg\max_t \hat{Y}_x(t)$. Building on this, we implement a **Causal-CARROT** variant by adapting both the parametric and kNN instantiations of CARROT to the R&C framework. We additionally include **CF-Regression**, which models $f : \mathcal{X} \to \mathbb{R}^{|\mathcal{T}|}$ and is trained to minimize MSE against the counterfactual utility function from the doubly robust estimator: $\min_f \sum_{i=1}^n \sum_{t=1}^{|\mathcal{T}|} (\hat{Y}_{x_i}(t) - f(x_i)_t)^2$. Decisions are made by selecting the treatment with the highest predicted value, i.e., $\hat{t} = \arg\max_t f(x)_t$.

Finally, we evaluate our regret-minimization methods. **RM-Classification** formulates the task as multi-class prediction over optimal treatments, serving as a classification-based upper bound. **RM-Softmax** directly minimizes a softmax-weighted regret surrogate. In the heterogeneous preference setting, we also assess **RM-Interval** (Section 3), which generalizes across a continuum of cost sensitivities by interpolating between models trained at discrete $\lambda$ values.

Table 1: Comparison of routing methods by causal reasoning and end-to-end training.

| Method | Causal | End-to-End |
|---|---|---|
| Full-Feedback | ✓ | ✓ |
| Baseline | ✗ | ✗ |
| R&C | ✓ | ✗ |
| Causal-CARROT (kNN & Embed) | ✓ | ✗ |
| CF-Regression | ✓ | ✗ |
| RM-Classification (Ours) | ✓ | ✓ |
| RM-Softmax (Ours) | ✓ | ✓ |
| RM-Interval (Ours) | ✓ | ✓ |

## 4.4 Evaluations

We evaluate our methods in two settings. In the $\lambda$-specific setting, each cost sensitivity value $\lambda \in \{0, 100, \ldots, 1000\}$ defines a separate routing task, with models trained and evaluated independently. In the heterogeneous preference setting, **RM-Interval** is trained on a subset $\{0, 200, \ldots, 1000\}$ and tested on held-out values $\{100, 300, 500, 700, 900\}$ to assess generalization across cost sensitivities.

We report the average utility across 10 independent trials for each routing method for SPROUT dataset and BERT embeddings in Table 2, where each trial involves randomly sampling observational data and retraining all models. In Figure 2, we also visualize the corresponding accuracy–cost curve associated with each method. Additional plots for the rest of the datasets and embeddings, along with detailed router performance are provided in Appendix B.

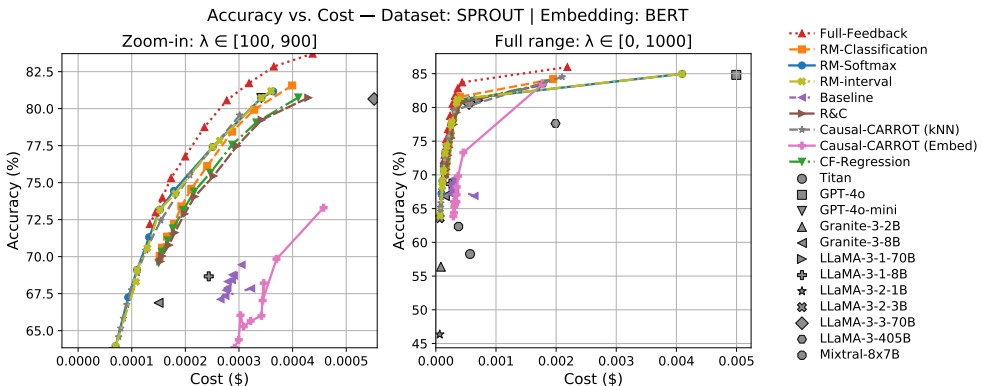

Figure 2: Accuracy–cost trade-off curve for SPROUT with BERT embeddings.

Table 2: Utility on **SPROUT** with BERT embeddings. *Full Feedback* serves as an oracle upper bound. The best-performing method for each column is highlighted in bold.

| Method | $\lambda = 0$ | $\lambda = 100$ | $\lambda = 200$ | $\lambda = 300$ | $\lambda = 400$ | $\lambda = 500$ |
|---|---|---|---|---|---|---|
| Full-Feedback | $85.99_{\pm0.17}$ | $79.34_{\pm0.13}$ | $75.55_{\pm0.29}$ | $72.16_{\pm0.24}$ | $69.47_{\pm0.23}$ | $66.97_{\pm0.37}$ |
| Baseline | $66.88_{\pm1.55}$ | $64.62_{\pm0.66}$ | $61.85_{\pm0.78}$ | $59.15_{\pm0.84}$ | $56.73_{\pm0.67}$ | $54.08_{\pm0.57}$ |
| R&C | $83.34_{\pm0.32}$ | $76.45_{\pm0.53}$ | $72.40_{\pm0.56}$ | $68.59_{\pm0.69}$ | $65.34_{\pm0.66}$ | $63.19_{\pm0.83}$ |
| Causal-CARROT (kNN) | $84.52_{\pm0.35}$ | $76.55_{\pm0.37}$ | $71.34_{\pm0.23}$ | $67.82_{\pm0.43}$ | $65.38_{\pm0.40}$ | $63.38_{\pm0.43}$ |
| Causal-CARROT (EmbedNet) | $83.46_{\pm0.39}$ | $68.73_{\pm0.74}$ | $62.44_{\pm2.40}$ | $56.72_{\pm1.68}$ | $54.37_{\pm2.12}$ | $48.91_{\pm2.41}$ |
| CF-Regression | $83.54_{\pm0.25}$ | $76.65_{\pm0.46}$ | $72.42_{\pm0.53}$ | $68.95_{\pm0.60}$ | $65.81_{\pm0.67}$ | $63.58_{\pm0.67}$ |
| RM-Classification | $84.20_{\pm0.24}$ | $\mathbf{77.58}_{\pm0.34}$ | $73.36_{\pm0.49}$ | $69.84_{\pm0.48}$ | $66.49_{\pm0.63}$ | $64.03_{\pm1.02}$ |
| RM-Softmax | $\mathbf{84.97}_{\pm0.39}$ | $77.53_{\pm0.81}$ | $\mathbf{73.89}_{\pm0.00}$ | $\mathbf{70.47}_{\pm0.00}$ | $\mathbf{67.38}_{\pm0.47}$ | $\mathbf{65.51}_{\pm0.60}$ |
| RM-Interval | $\mathbf{84.97}_{\pm0.39}$ | $77.60_{\pm0.62}$ | $\mathbf{73.89}_{\pm0.00}$ | $69.92_{\pm0.57}$ | $\mathbf{67.38}_{\pm0.47}$ | $65.20_{\pm0.67}$ |

| Method | $\lambda = 600$ | $\lambda = 700$ | $\lambda = 800$ | $\lambda = 900$ | $\lambda = 1000$ | |
|---|---|---|---|---|---|---|
| Full-Feedback | $64.79_{\pm0.31}$ | $63.18_{\pm0.21}$ | $61.43_{\pm0.27}$ | $59.98_{\pm0.30}$ | $58.83_{\pm0.24}$ | |
| Baseline | $51.17_{\pm0.54}$ | $48.79_{\pm0.67}$ | $45.74_{\pm1.21}$ | $42.58_{\pm1.36}$ | $38.97_{\pm2.65}$ | |
| R&C | $60.98_{\pm0.81}$ | $59.01_{\pm0.86}$ | $57.21_{\pm0.99}$ | $55.97_{\pm1.11}$ | $54.33_{\pm1.37}$ | |
| Causal-CARROT (kNN) | $61.77_{\pm0.43}$ | $60.39_{\pm0.38}$ | $59.07_{\pm0.37}$ | $57.99_{\pm0.35}$ | $56.99_{\pm0.34}$ | |
| Causal-CARROT (EmbedNet) | $46.35_{\pm1.53}$ | $43.66_{\pm3.04}$ | $41.81_{\pm2.34}$ | $37.42_{\pm5.86}$ | $34.69_{\pm4.44}$ | |
| CF-Regression | $61.37_{\pm0.64}$ | $59.52_{\pm0.54}$ | $57.72_{\pm0.72}$ | $56.31_{\pm0.66}$ | $54.56_{\pm0.71}$ | |
| RM-Classification | $61.84_{\pm1.13}$ | $59.68_{\pm1.48}$ | $58.09_{\pm1.22}$ | $56.52_{\pm1.63}$ | $54.85_{\pm1.76}$ | |
| RM-Softmax | $\mathbf{64.03}_{\pm0.39}$ | $\mathbf{62.04}_{\pm1.17}$ | $\mathbf{60.32}_{\pm1.31}$ | $\mathbf{58.85}_{\pm1.13}$ | $\mathbf{56.95}_{\pm0.55}$ | |
| RM-Interval | $\mathbf{64.03}_{\pm0.39}$ | $61.54_{\pm1.31}$ | $\mathbf{60.32}_{\pm1.31}$ | $58.58_{\pm1.15}$ | $\mathbf{56.95}_{\pm0.55}$ | |

Our RM-based approaches consistently deliver the strongest performance overall, with **RM-Softmax** and **RM-Interval** standing out for both high utility and low variance. Notably, **RM-Interval** generalizes remarkably well to unseen budget levels (i.e., odd $\lambda$ values), many times even outperforming models trained specifically on those points. These results underscore the effectiveness of our regret-minimization framework in both fixed and variable cost settings.

The standard **Baseline** method, which reflects the common decoupled approach used in prior work and ignores treatment selection bias, performs the worst across most values of $\lambda$, underscoring the importance of accounting for treatment bias in observational data. Notably, the simple **R&C** method and the **Causal-CARROT** variants (which incorporate causal corrections) achieve substantial improvements over it, validating our claim that bias-aware routing significantly improves performance.

The performance gap between **CF-Regression** and our RM-based methods demonstrates the benefit of an integrated end-to-end approach. Whereas **CF-Regression** focuses on approximating the counterfactual utility function and then selecting the best model based on predicted outcomes, our methods directly minimize regret, leading to superior and more stable results. Comparing our two surrogate formulations, **RM-Softmax** generally outperforms **RM-Classification** in both utility and variance, indicating the advantage of optimizing a differentiable surrogate objective. Finally, among the **Causal-CARROT** variants, the kNN version consistently outperforms the EmbedNet variant, suggesting that non-parametric estimators may offer greater robustness in this setting.

# 5 Conclusion

We propose a causal end-to-end framework for routing queries to LLMs under observational data. Our approach introduces a regret-minimizing objective grounded in counterfactual estimation, enabling principled policy learning that accounts for treatment selection bias without requiring full-feedback data. Unlike prior approaches that rely on decoupled prediction of accuracy and cost, where errors can compound, our method directly optimizes the decision objective. To support heterogeneous user preferences, we develop an interval-conditioned routing architecture that generalizes across a continuum of cost-sensitivity parameters. Theoretical analysis provides guarantees on interpolation sufficiency and regret bounds, while empirical evaluations on public routing benchmarks demonstrate that our methods consistently outperform strong baselines, including recent routing algorithms, across multiple embedding models. Future work includes extending the framework to accommodate additional user-defined metrics or hard constraints that cannot be readily incorporated as soft penalties in the objective. Another promising direction is to explore online or adaptive routing in dynamic environments, as well as extending causal regret minimization to multi-turn settings.

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

# A Proofs of Propositions

*Proposition* 1. Suppose the estimated utility function $\hat{Y}_x : \mathcal{T} \to \mathbb{R}$ is $L$-Lipschitz continuous over the probability simplex with respect to the $\ell_1$ norm, as in Definition 1. Then, for a policy $f : \mathcal{X} \to \Delta^{|\mathcal{T}|}$ that outputs a distribution $f(x)$ over $\mathcal{T}$, the regret can be upper bounded by:

$$\text{Regret}(f) \leq L \cdot \frac{1}{n} \sum_{i=1}^{n} \sqrt{2 \cdot \text{CE}(t_i^*, f(x_i))}, \tag{15}$$

where $t_i^* := \arg\max_{t \in \mathcal{T}} \hat{Y}_{x_i}(t)$ is the optimal treatment for input $x_i$, and $\text{CE}(t_i^*, f(x_i)) := -\log f(x_i)_{t_i^*}$ denotes the cross-entropy loss.

*Proof.* Let $e_{t_i^*} \in \Delta^{|\mathcal{T}|}$ denote the one-hot distribution over the optimal treatment $t_i^*$. By the definition of regret:

$$\text{Regret}(f) = \frac{1}{n} \sum_{i=1}^{n} \left[ \hat{Y}_{x_i}(e_{t_i^*}) - \hat{Y}_{x_i}(f(x_i)) \right]. \tag{16}$$

Using the $L$-Lipschitz continuity of $\hat{Y}_{x_i}(\cdot)$ under the $\ell_1$ norm:

$$\left| \hat{Y}_{x_i}(e_{t_i^*}) - \hat{Y}_{x_i}(f(x_i)) \right| \leq L \cdot \| e_{t_i^*} - f(x_i) \|_1. \tag{17}$$

Applying Pinsker's inequality:

$$\| e_{t_i^*} - f(x_i) \|_1 \leq \sqrt{2 \cdot \text{KL}(e_{t_i^*} \parallel f(x_i))} = \sqrt{2 \cdot \text{CE}(t_i^*, f(x_i))}, \tag{18}$$

where the equality follows because KL divergence from a one-hot distribution to a probability vector reduces to cross-entropy. Combining the above:

$$\text{Regret}(f) \leq L \cdot \frac{1}{n} \sum_{i=1}^{n} \sqrt{2 \cdot \text{CE}(t_i^*, f(x_i))}, \tag{19}$$

which completes the proof. $\square$

*Proposition* 2. Let $f : \mathcal{X} \to \mathbb{R}^{|\mathcal{T}|}$ be a neural network whose output is passed through a softmax layer with fixed temperature $\tau > 0$, and define $t_i^* := \arg\max_{t \in \mathcal{T}} \hat{Y}_{x_i}(t)$. Then, optimizing the following objective using gradient descent

$$\min_{f} \frac{1}{n} \sum_{i=1}^{n} \left( \hat{Y}_{x_i}(t_i^*) - \sum_{t=1}^{|\mathcal{T}|} \hat{Y}_{x_i}(t) \cdot \text{softmax}(f(x_i))_t \right) \tag{20}$$

leads the model $f$ to place all probability mass on the optimal treatment $t_i^*$. That is, at convergence,

$$\text{softmax}(f(x_i))_t \to \begin{cases} 1 & \text{if } t = t_i^*, \\ 0 & \text{otherwise.} \end{cases} \tag{21}$$

*Proof.* Let $\hat{Y}_{x_i} \in \mathbb{R}^{|\mathcal{T}|}$ denote the vector of estimated potential outcomes for input $x_i$, and let $f(x_i) \in \mathbb{R}^{|\mathcal{T}|}$ be the output of the neural network before the softmax layer. The objective for a single instance $x_i$ can be written as minimizing the regret surrogate:

$$\hat{Y}_{x_i}(t_i^*) - \sum_{t=1}^{|\mathcal{T}|} \hat{Y}_{x_i}(t) \cdot \text{softmax}(f(x_i))_t. \tag{22}$$

This is equivalent to maximizing the inner product:

$$\langle \hat{Y}_{x_i}, \text{softmax}(f(x_i)) \rangle. \tag{23}$$

Let us denote $p := \text{softmax}(f(x_i)) \in \Delta^{|\mathcal{T}|-1}$, the probability simplex. We now show that the inner product $\langle \hat{Y}_{x_i}, p \rangle$ increases at each gradient step. Since $p = \text{softmax}(f(x_i))$, we can compute the gradient of the objective with respect to $f(x_i)$ as:

$$\nabla_{f(x_i)} \langle \hat{Y}_{x_i}, \text{softmax}(f(x_i)) \rangle = J_{\text{softmax}}(f(x_i))^{\top} \hat{Y}_{x_i}, \tag{24}$$

where $J_{\text{softmax}}(f(x_i))$ is the Jacobian of the softmax function, given by:

$$J_{\text{softmax}}(f(x_i))_{t,s} = \frac{\partial \, \text{softmax}(f(x_i))_t}{\partial f(x_i)_s} = \text{softmax}(f(x_i))_t \left( \delta_{t,s} - \text{softmax}(f(x_i))_s \right). \tag{25}$$

This gradient direction corresponds to increasing the logit value of actions with higher $\hat{Y}_{x_i}(t)$ and decreasing those with lower values, pushing the softmax distribution toward the mode of $\hat{Y}_{x_i}$. In other words, the gradient ascent step increases the inner product at each iteration $k$:

$$\langle \hat{Y}_{x_i}, \text{softmax}(f(x_i)) \rangle^{(k+1)} > \langle \hat{Y}_{x_i}, \text{softmax}(f(x_i)) \rangle^{(k)}. \tag{26}$$

Since $\hat{Y}_{x_i}$ is fixed and the softmax is smooth and bounded, this sequence is monotonically increasing and converges to the maximum possible value:

$$\langle \hat{Y}_{x_i}, \text{softmax}(f(x_i)) \rangle \to \max_t \hat{Y}_{x_i}(t) = \hat{Y}_{x_i}(t_i^*), \tag{27}$$

which implies:

$$\text{softmax}(f(x_i))_t \to \begin{cases} 1 & \text{if } t = t_i^*, \\ 0 & \text{otherwise.} \end{cases} \tag{28}$$

Thus, the regret surrogate converges to zero:

$$\hat{Y}_{x_i}(t_i^*) - \langle \hat{Y}_{x_i}, \text{softmax}(f(x_i)) \rangle \to 0, \tag{29}$$

and the learned policy selects the treatment maximizing the estimated outcome. $\square$

*Proposition* 3 (Piecewise Constant Optimal Policy). Fix a query $x \in \mathcal{X}$ and assume the estimated utility function is affine in $\lambda$, i.e., $\hat{Y}_x^{\lambda}(t) = a_x(t) - \lambda \cdot c_x(t)$ for all $t \in \mathcal{T}$. Then the optimal treatment

$$t^*(\lambda) := \arg\max_{t \in \mathcal{T}} \hat{Y}_x^{\lambda}(t)$$

is piecewise constant in $\lambda$. That is, the budget space $\mathbb{R}_{\geq 0}$ can be partitioned into intervals over which the optimal treatment remains fixed.

*Proof.* For fixed $x$, each $\hat{Y}_x^{\lambda}(t)$ is an affine function of $\lambda$. The pointwise maximum of a finite collection of affine functions is piecewise affine, and the argmax corresponds to the highest line at each $\lambda$. Since each pair of lines can intersect at most once, the number of intervals over which a single treatment is optimal is bounded by $|\mathcal{T}| - 1$. Therefore, $t^*(\lambda)$ changes only at these intersection points and remains constant within each interval. $\square$

*Proposition* 4 (Affine Closure of Utility Function). Let $\lambda_j < \lambda_{j+1}$ be two adjacent budget values and let $\lambda \in [\lambda_j, \lambda_{j+1}]$. Suppose the utility function is affine in $\lambda$:

$$\hat{Y}_x^{\lambda}(t) = a_x(t) - \lambda \cdot c_x(t).$$

Then for all $t \in \mathcal{T}$, the utility at $\lambda$ is a convex combination of utilities at the endpoints:

$$\hat{Y}_x^{\lambda}(t) = \alpha \cdot \hat{Y}_x^{\lambda_j}(t) + (1 - \alpha) \cdot \hat{Y}_x^{\lambda_{j+1}}(t), \quad \text{where } \alpha := \frac{\lambda_{j+1} - \lambda}{\lambda_{j+1} - \lambda_j}.$$

*Proof.* We expand each term:

$$\hat{Y}_x^{\lambda_j}(t) = a_x(t) - \lambda_j \cdot c_x(t),$$
$$\hat{Y}_x^{\lambda_{j+1}}(t) = a_x(t) - \lambda_{j+1} \cdot c_x(t).$$

Then:

$$\alpha \cdot \hat{Y}_x^{\lambda_j}(t) + (1-\alpha) \cdot \hat{Y}_x^{\lambda_{j+1}}(t) = \alpha \cdot (a_x(t) - \lambda_j c_x(t)) + (1-\alpha) \cdot (a_x(t) - \lambda_{j+1} c_x(t))$$
$$= a_x(t) - [\alpha\lambda_j + (1-\alpha)\lambda_{j+1}] \cdot c_x(t)$$
$$= a_x(t) - \lambda \cdot c_x(t) = \hat{Y}_x^{\lambda}(t),$$

since:

$$\alpha\lambda_j + (1-\alpha)\lambda_{j+1} = \lambda.$$

$\square$

*Corollary* 1 (Sufficiency of Two Models per Interval). Under the affine assumption, the utility $\hat{Y}_x^{\lambda}(t)$ for any $\lambda \in [\lambda_j, \lambda_{j+1}]$ can be exactly reconstructed using only the endpoints $\hat{Y}_x^{\lambda_j}(t)$ and $\hat{Y}_x^{\lambda_{j+1}}(t)$. Thus, it is sufficient to use only the two corresponding models $f_{\lambda_j}$ and $f_{\lambda_{j+1}}$ for interpolation within the interval.

*Proof.* This follows immediately from the statement of Proposition 4. $\square$

*Proposition* 5 (Expressivity of Additive Two-Model joint Architecture). Let $\lambda \in [\lambda_j, \lambda_{j+1}]$, and suppose that for each $t \in \mathcal{T}$ the utility function satisfies $\hat{Y}_x^{\lambda}(t) = a_x(t) - \lambda \cdot c_x(t)$. Then the optimal treatment $t^*(\lambda) := \arg\max_t \hat{Y}_x^{\lambda}(t)$ can be exactly represented by a softmax policy over a function of the form:

$$f(x, \lambda) = \texttt{Linear}\left([f_{\lambda_j}(x), f_{\lambda_{j+1}}(x)] + g(\lambda)\right),$$

where $g(\lambda)$ is any differentiable embedding of $\lambda$, and $f_{\lambda_j}, f_{\lambda_{j+1}}$ are accurate predictors trained at endpoints $\lambda_j$ and $\lambda_{j+1}$.

*Proof.* From Proposition 4, the utility $\hat{Y}_x^{\lambda}(t)$ is a convex combination of $\hat{Y}_x^{\lambda_j}(t)$ and $\hat{Y}_x^{\lambda_{j+1}}(t)$. If the network $f(x, \lambda)$ linearly combines the outputs of $f_{\lambda_j}(x)$ and $f_{\lambda_{j+1}}(x)$, then its scores can match $\hat{Y}_x^{\lambda}(t)$ up to a scalar transformation. Applying softmax preserves the argmax.

Including $g(\lambda)$ allows the architecture to learn any additional monotonic reweighting of the interpolation, ensuring the output scores can be shaped to approximate the true utility surface exactly. Thus, the architecture can represent the optimal policy within each interval. $\square$

# B   Additional Results

In this section, we present the additional plots for the rest of the datasets as well as the exact values of utility for value $\lambda$. We begin by presenting the rest of the figures.

## B.1   Additional Figures

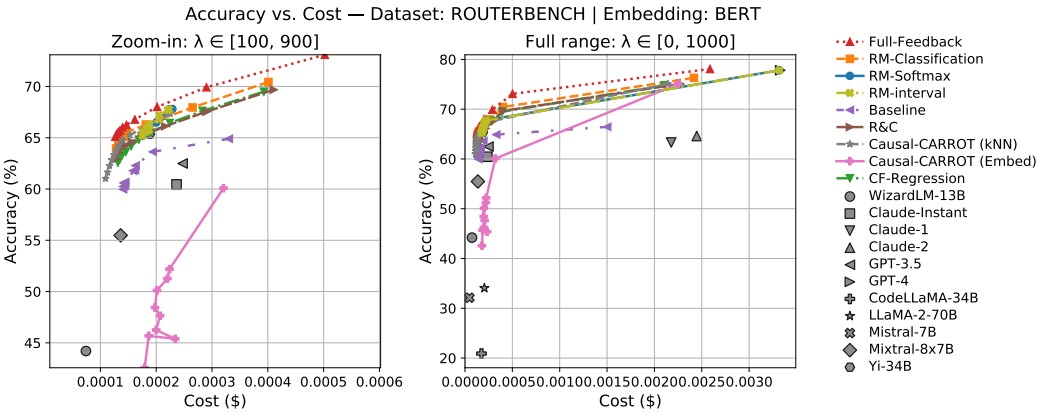

Figure 3: Accuracy–cost trade-off curve for RouterBench with BERT embeddings.

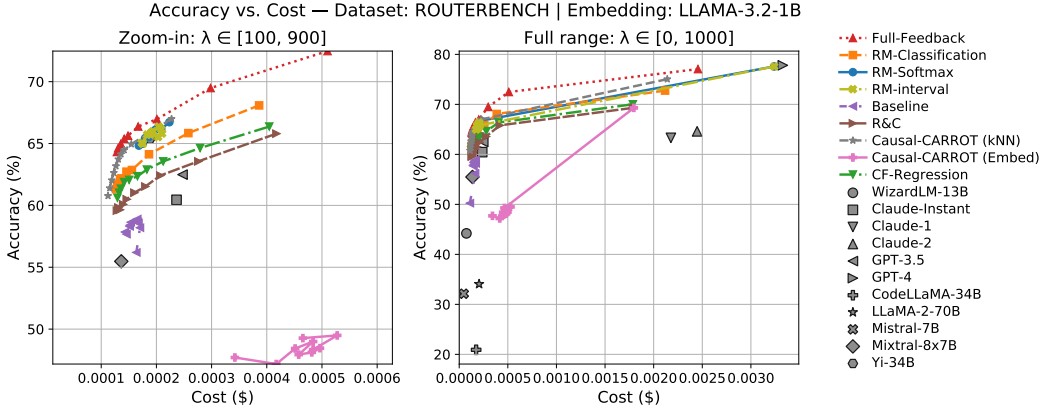

Figure 4: Accuracy–cost trade-off curve for RouterBench with LLaMa-3.2-1B embeddings.

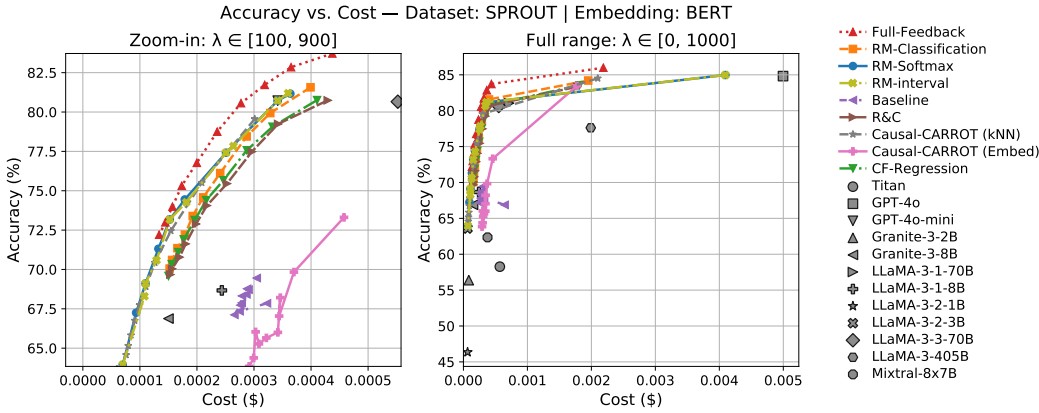

Figure 5: Accuracy–cost trade-off curve for SPROUT with BERT embeddings.

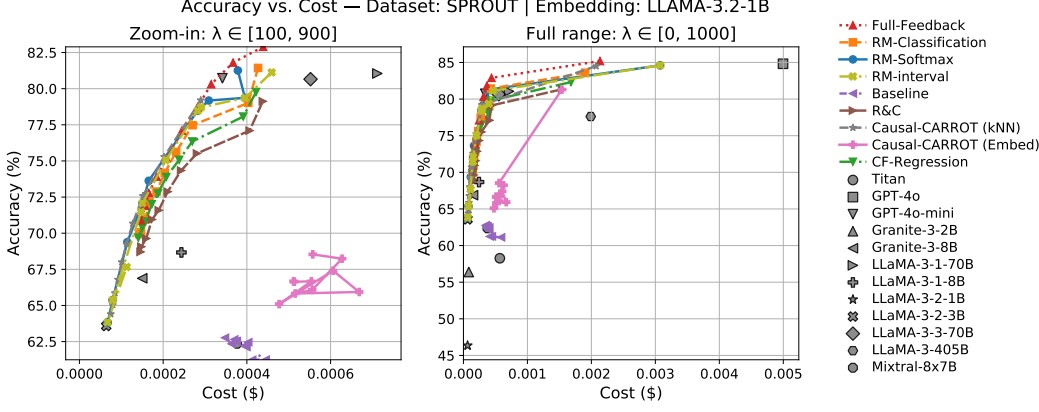

Figure 6: Accuracy–cost trade-off curve for SPROUT with LLaMa-3.2-1B embeddings.

## B.2  Additional Tables

Table 3: Utility for **RouterBench** with `BERT` embeddings. *Full Feedback* serves as an oracle upper bound. The best-performing method for each column is highlighted in bold.

| Method | $\lambda = 0$ | $\lambda = 100$ | $\lambda = 200$ | $\lambda = 300$ | $\lambda = 400$ | $\lambda = 500$ | $\lambda = 600$ | $\lambda = 700$ | $\lambda = 800$ | $\lambda = 900$ | $\lambda = 1000$ |
|---|---|---|---|---|---|---|---|---|---|---|---|
| Full-Feedback | $78.07_{\pm0.14}$ | $68.07_{\pm0.24}$ | $64.12_{\pm0.18}$ | $61.96_{\pm0.13}$ | $60.31_{\pm0.13}$ | $58.92_{\pm0.18}$ | $57.62_{\pm0.21}$ | $56.26_{\pm0.30}$ | $55.00_{\pm0.24}$ | $53.73_{\pm0.31}$ | $52.42_{\pm0.18}$ |
| Baseline | $66.46_{\pm0.66}$ | $61.60_{\pm0.64}$ | $59.75_{\pm0.46}$ | $57.41_{\pm0.49}$ | $55.37_{\pm0.51}$ | $53.85_{\pm0.96}$ | $51.67_{\pm1.04}$ | $50.55_{\pm1.07}$ | $48.72_{\pm1.28}$ | $47.87_{\pm0.76}$ | $46.19_{\pm0.46}$ |
| R&C | $75.08_{\pm0.49}$ | $65.58_{\pm0.28}$ | $61.73_{\pm0.53}$ | $59.63_{\pm0.53}$ | $58.23_{\pm0.58}$ | $56.79_{\pm0.63}$ | $55.47_{\pm0.64}$ | $54.34_{\pm0.46}$ | $53.08_{\pm0.58}$ | $51.65_{\pm0.59}$ | $50.18_{\pm0.53}$ |
| Causal-CARROT (kNN) | $75.12_{\pm0.64}$ | $65.12_{\pm0.51}$ | $62.21_{\pm0.52}$ | $60.56_{\pm0.49}$ | $58.94_{\pm0.41}$ | $57.39_{\pm0.42}$ | $55.86_{\pm0.35}$ | $54.41_{\pm0.42}$ | $52.95_{\pm0.46}$ | $51.47_{\pm0.49}$ | $50.08_{\pm0.60}$ |
| Causal-CARROT (EmbedNet) | $75.07_{\pm0.50}$ | $56.87_{\pm1.39}$ | $47.71_{\pm4.76}$ | $44.66_{\pm1.95}$ | $42.06_{\pm2.31}$ | $38.55_{\pm3.30}$ | $35.22_{\pm2.27}$ | $32.25_{\pm2.40}$ | $26.66_{\pm6.05}$ | $28.89_{\pm3.53}$ | $24.64_{\pm5.52}$ |
| CF-Regression | $74.81_{\pm0.55}$ | $65.56_{\pm0.17}$ | $61.95_{\pm0.44}$ | $59.71_{\pm0.69}$ | $57.91_{\pm0.65}$ | $56.36_{\pm0.73}$ | $54.87_{\pm0.55}$ | $53.44_{\pm0.63}$ | $52.19_{\pm0.48}$ | $50.78_{\pm0.51}$ | $49.43_{\pm0.39}$ |
| RM-Classification | $76.30_{\pm0.61}$ | $\mathbf{66.43}_{\pm0.28}$ | $62.65_{\pm0.64}$ | $60.85_{\pm0.54}$ | $\mathbf{59.65}_{\pm0.51}$ | $\mathbf{58.07}_{\pm0.62}$ | $\mathbf{56.80}_{\pm0.47}$ | $\mathbf{55.29}_{\pm0.58}$ | $\mathbf{54.07}_{\pm0.62}$ | $\mathbf{52.45}_{\pm0.72}$ | $\mathbf{51.14}_{\pm0.61}$ |
| RM-Softmax | $\mathbf{77.82}_{\pm0.03}$ | $65.51_{\pm0.51}$ | $\mathbf{63.30}_{\pm0.26}$ | $60.75_{\pm0.61}$ | $58.58_{\pm0.59}$ | $56.48_{\pm0.52}$ | $54.53_{\pm0.57}$ | $52.79_{\pm0.74}$ | $51.10_{\pm0.98}$ | $49.22_{\pm1.14}$ | $47.64_{\pm1.40}$ |
| RM-Interval | $\mathbf{77.82}_{\pm0.03}$ | $65.51_{\pm0.30}$ | $\mathbf{63.30}_{\pm0.26}$ | $\mathbf{61.01}_{\pm0.51}$ | $58.58_{\pm0.59}$ | $56.92_{\pm0.31}$ | $54.53_{\pm0.57}$ | $53.11_{\pm0.77}$ | $51.10_{\pm0.98}$ | $49.60_{\pm1.21}$ | $47.64_{\pm1.40}$ |

Table 4: Utility for **RouterBench** with `Llama-3.2-1B` embeddings. The best-performing method for each column is highlighted in bold.

| Method | $\lambda = 0$ | $\lambda = 100$ | $\lambda = 200$ | $\lambda = 300$ | $\lambda = 400$ | $\lambda = 500$ | $\lambda = 600$ | $\lambda = 700$ | $\lambda = 800$ | $\lambda = 900$ | $\lambda = 1000$ |
|---|---|---|---|---|---|---|---|---|---|---|---|
| Full-Feedback | $77.06_{\pm0.30}$ | $67.37_{\pm0.35}$ | $63.52_{\pm0.35}$ | $60.94_{\pm0.27}$ | $59.70_{\pm0.24}$ | $58.20_{\pm0.19}$ | $56.88_{\pm0.33}$ | $55.49_{\pm0.38}$ | $54.03_{\pm0.14}$ | $52.86_{\pm0.25}$ | $51.47_{\pm0.28}$ |
| Baseline | $50.24_{\pm1.28}$ | $54.56_{\pm0.96}$ | $54.79_{\pm1.11}$ | $53.54_{\pm0.63}$ | $52.00_{\pm0.57}$ | $50.63_{\pm0.68}$ | $49.43_{\pm0.66}$ | $47.78_{\pm1.22}$ | $46.19_{\pm0.61}$ | $44.96_{\pm0.66}$ | $43.10_{\pm0.84}$ |
| R&C | $69.30_{\pm0.63}$ | $61.62_{\pm0.70}$ | $58.04_{\pm0.44}$ | $56.20_{\pm0.52}$ | $54.33_{\pm0.58}$ | $53.02_{\pm0.62}$ | $51.72_{\pm0.57}$ | $50.30_{\pm0.50}$ | $48.94_{\pm0.48}$ | $48.03_{\pm0.56}$ | $46.73_{\pm0.56}$ |
| Causal-CARROT (kNN) | $75.04_{\pm0.47}$ | $64.72_{\pm0.47}$ | $61.86_{\pm0.53}$ | $\mathbf{60.23}_{\pm0.54}$ | $\mathbf{58.71}_{\pm0.49}$ | $\mathbf{57.18}_{\pm0.48}$ | $\mathbf{55.58}_{\pm0.50}$ | $\mathbf{54.04}_{\pm0.49}$ | $\mathbf{52.50}_{\pm0.51}$ | $\mathbf{51.00}_{\pm0.59}$ | $\mathbf{49.54}_{\pm0.54}$ |
| Causal-CARROT (EmbedNet) | $69.30_{\pm0.38}$ | $44.62_{\pm1.38}$ | $38.94_{\pm2.63}$ | $33.82_{\pm2.07}$ | $28.60_{\pm4.29}$ | $24.04_{\pm3.76}$ | $20.47_{\pm6.44}$ | $15.11_{\pm3.78}$ | $12.32_{\pm5.07}$ | $9.64_{\pm6.26}$ | $13.48_{\pm1.45}$ |
| CF-Regression | $70.02_{\pm0.33}$ | $62.33_{\pm0.78}$ | $59.05_{\pm0.24}$ | $57.19_{\pm0.42}$ | $55.58_{\pm0.56}$ | $54.08_{\pm0.43}$ | $53.03_{\pm0.64}$ | $51.97_{\pm0.64}$ | $50.50_{\pm0.56}$ | $49.10_{\pm0.45}$ | $47.68_{\pm0.44}$ |
| RM-Classification | $72.76_{\pm0.50}$ | $64.22_{\pm0.65}$ | $60.68_{\pm0.56}$ | $58.54_{\pm0.46}$ | $56.64_{\pm0.72}$ | $55.44_{\pm0.33}$ | $53.98_{\pm0.46}$ | $52.71_{\pm0.44}$ | $51.14_{\pm0.59}$ | $50.06_{\pm0.55}$ | $48.63_{\pm0.36}$ |
| RM-Softmax | $\mathbf{77.58}_{\pm0.49}$ | $\mathbf{64.49}_{\pm0.50}$ | $\mathbf{61.97}_{\pm0.52}$ | $60.17_{\pm0.61}$ | $58.20_{\pm0.46}$ | $56.21_{\pm0.40}$ | $54.34_{\pm0.51}$ | $52.86_{\pm0.79}$ | $50.96_{\pm0.92}$ | $49.70_{\pm1.19}$ | $47.69_{\pm1.65}$ |
| RM-Interval | $\mathbf{77.58}_{\pm0.49}$ | $63.70_{\pm1.47}$ | $\mathbf{61.97}_{\pm0.52}$ | $59.43_{\pm1.56}$ | $58.20_{\pm0.46}$ | $56.31_{\pm0.47}$ | $54.34_{\pm0.51}$ | $52.55_{\pm0.91}$ | $50.96_{\pm0.92}$ | $49.23_{\pm1.81}$ | $47.69_{\pm1.65}$ |

Table 5: Utility for **SPOUT** with `BERT` embeddings. The best-performing method for each column is highlighted in bold.

| Method | $\lambda = 0$ | $\lambda = 100$ | $\lambda = 200$ | $\lambda = 300$ | $\lambda = 400$ | $\lambda = 500$ | $\lambda = 600$ | $\lambda = 700$ | $\lambda = 800$ | $\lambda = 900$ | $\lambda = 1000$ |
|---|---|---|---|---|---|---|---|---|---|---|---|
| Full-Feedback | $85.99_{\pm0.17}$ | $79.34_{\pm0.13}$ | $75.55_{\pm0.29}$ | $72.16_{\pm0.24}$ | $69.47_{\pm0.23}$ | $66.97_{\pm0.37}$ | $64.79_{\pm0.31}$ | $63.18_{\pm0.21}$ | $61.43_{\pm0.27}$ | $59.98_{\pm0.30}$ | $58.83_{\pm0.24}$ |
| Baseline | $66.88_{\pm1.55}$ | $64.62_{\pm0.66}$ | $61.85_{\pm0.78}$ | $59.15_{\pm0.84}$ | $56.73_{\pm0.67}$ | $54.08_{\pm0.57}$ | $51.17_{\pm0.54}$ | $48.79_{\pm0.67}$ | $45.74_{\pm1.21}$ | $42.58_{\pm1.36}$ | $38.97_{\pm2.65}$ |
| R&C | $83.34_{\pm1.32}$ | $76.45_{\pm0.52}$ | $72.40_{\pm0.56}$ | $68.59_{\pm0.69}$ | $65.34_{\pm0.66}$ | $63.19_{\pm0.83}$ | $60.98_{\pm0.81}$ | $59.01_{\pm0.86}$ | $57.21_{\pm0.99}$ | $55.97_{\pm1.11}$ | $54.33_{\pm1.37}$ |
| Causal-CARROT (kNN) | $84.52_{\pm0.35}$ | $76.55_{\pm0.37}$ | $71.34_{\pm0.23}$ | $67.82_{\pm0.43}$ | $65.38_{\pm0.40}$ | $63.38_{\pm0.43}$ | $61.77_{\pm0.43}$ | $60.39_{\pm0.38}$ | $59.07_{\pm0.37}$ | $57.99_{\pm0.35}$ | $56.99_{\pm0.34}$ |
| Causal-CARROT (EmbedNet) | $83.46_{\pm0.39}$ | $68.73_{\pm0.74}$ | $62.44_{\pm2.40}$ | $56.72_{\pm1.68}$ | $54.37_{\pm2.12}$ | $48.91_{\pm2.41}$ | $46.35_{\pm1.53}$ | $43.66_{\pm3.04}$ | $41.81_{\pm2.34}$ | $37.42_{\pm5.86}$ | $34.69_{\pm4.44}$ |
| CF-Regression | $83.54_{\pm0.25}$ | $76.65_{\pm0.46}$ | $72.42_{\pm0.53}$ | $68.95_{\pm0.60}$ | $65.81_{\pm0.67}$ | $63.58_{\pm0.67}$ | $61.37_{\pm0.64}$ | $59.52_{\pm0.54}$ | $57.72_{\pm0.72}$ | $56.31_{\pm0.66}$ | $54.56_{\pm0.71}$ |
| RM-Classification | $84.20_{\pm0.24}$ | $\mathbf{77.58}_{\pm0.34}$ | $73.36_{\pm0.49}$ | $69.84_{\pm0.48}$ | $66.49_{\pm0.63}$ | $64.03_{\pm1.02}$ | $61.84_{\pm1.13}$ | $59.68_{\pm1.48}$ | $58.09_{\pm1.22}$ | $56.52_{\pm1.63}$ | $54.85_{\pm1.76}$ |
| RM-Softmax | $\mathbf{84.97}_{\pm0.39}$ | $77.53_{\pm0.81}$ | $\mathbf{73.89}_{\pm0.00}$ | $\mathbf{70.47}_{\pm0.00}$ | $\mathbf{67.38}_{\pm0.47}$ | $\mathbf{65.51}_{\pm0.60}$ | $\mathbf{64.03}_{\pm0.39}$ | $\mathbf{62.04}_{\pm1.17}$ | $\mathbf{60.32}_{\pm1.31}$ | $\mathbf{58.85}_{\pm1.13}$ | $\mathbf{56.95}_{\pm0.55}$ |
| RM-Interval | $\mathbf{84.97}_{\pm0.39}$ | $77.60_{\pm0.62}$ | $\mathbf{73.89}_{\pm0.00}$ | $69.92_{\pm0.57}$ | $\mathbf{67.38}_{\pm0.47}$ | $65.20_{\pm0.67}$ | $\mathbf{64.03}_{\pm0.39}$ | $61.54_{\pm1.31}$ | $\mathbf{60.32}_{\pm1.31}$ | $58.58_{\pm1.15}$ | $\mathbf{56.95}_{\pm0.55}$ |

Table 6: Utility for **SPOUT** with `LLaMa-3.2-1B` embeddings. The best-performing method for each column is highlighted in bold.

| Method | $\lambda = 0$ | $\lambda = 100$ | $\lambda = 200$ | $\lambda = 300$ | $\lambda = 400$ | $\lambda = 500$ | $\lambda = 600$ | $\lambda = 700$ | $\lambda = 800$ | $\lambda = 900$ | $\lambda = 1000$ |
|---|---|---|---|---|---|---|---|---|---|---|---|
| Full-Feedback | $85.19_{\pm0.17}$ | $78.50_{\pm0.35}$ | $74.47_{\pm0.28}$ | $70.87_{\pm0.23}$ | $67.44_{\pm0.22}$ | $64.87_{\pm0.54}$ | $62.69_{\pm0.36}$ | $61.01_{\pm0.57}$ | $59.51_{\pm0.55}$ | $57.75_{\pm0.60}$ | $56.22_{\pm0.58}$ |
| Baseline | $61.12_{\pm1.95}$ | $57.12_{\pm1.66}$ | $52.35_{\pm2.63}$ | $51.23_{\pm1.04}$ | $47.86_{\pm2.19}$ | $42.50_{\pm2.64}$ | $40.05_{\pm4.79}$ | $34.20_{\pm5.47}$ | $30.20_{\pm5.82}$ | $29.64_{\pm5.25}$ | $27.99_{\pm5.66}$ |
| R&C | $81.34_{\pm0.41}$ | $74.75_{\pm0.54}$ | $68.97_{\pm5.81}$ | $67.09_{\pm0.67}$ | $64.62_{\pm0.57}$ | $62.33_{\pm0.63}$ | $60.22_{\pm0.46}$ | $58.81_{\pm0.48}$ | $56.86_{\pm0.81}$ | $55.76_{\pm1.17}$ | $54.10_{\pm1.48}$ |
| Causal-CARROT (kNN) | $84.50_{\pm0.28}$ | $76.30_{\pm0.42}$ | $71.24_{\pm0.67}$ | $68.02_{\pm0.50}$ | $65.57_{\pm0.53}$ | $63.62_{\pm0.47}$ | $61.93_{\pm0.43}$ | $60.33_{\pm0.41}$ | $\mathbf{59.05}_{\pm0.37}$ | $\mathbf{57.86}_{\pm0.35}$ | $56.89_{\pm0.29}$ |
| Causal-CARROT (EmbedNet) | $81.29_{\pm0.48}$ | $62.96_{\pm1.32}$ | $55.69_{\pm4.27}$ | $49.40_{\pm5.41}$ | $45.21_{\pm4.47}$ | $32.52_{\pm14.00}$ | $31.05_{\pm11.57}$ | $31.70_{\pm6.75}$ | $22.34_{\pm9.09}$ | $20.47_{\pm9.64}$ | $15.52_{\pm6.70}$ |
| CF-Regression | $82.32_{\pm0.45}$ | $75.55_{\pm0.43}$ | $70.24_{\pm5.37}$ | $68.26_{\pm0.64}$ | $65.55_{\pm0.58}$ | $63.51_{\pm0.52}$ | $61.49_{\pm0.53}$ | $59.90_{\pm0.60}$ | $58.19_{\pm0.60}$ | $56.86_{\pm0.61}$ | $55.61_{\pm0.76}$ |
| RM-Classification | $83.60_{\pm0.43}$ | $77.17_{\pm0.47}$ | $70.95_{\pm6.42}$ | $69.36_{\pm0.62}$ | $66.38_{\pm0.76}$ | $63.99_{\pm0.49}$ | $61.82_{\pm0.19}$ | $60.45_{\pm0.08}$ | $58.38_{\pm0.11}$ | $56.83_{\pm0.83}$ | $55.64_{\pm1.35}$ |
| RM-Softmax | $\mathbf{84.60}_{\pm0.89}$ | $\mathbf{77.48}_{\pm1.21}$ | $\mathbf{71.45}_{\pm6.01}$ | $69.91_{\pm0.85}$ | $\mathbf{67.18}_{\pm0.51}$ | $\mathbf{65.35}_{\pm0.60}$ | $\mathbf{63.04}_{\pm1.35}$ | $\mathbf{61.38}_{\pm1.61}$ | $59.04_{\pm1.05}$ | $57.72_{\pm0.33}$ | $\mathbf{57.12}_{\pm0.21}$ |
| RM-Interval | $\mathbf{84.60}_{\pm0.89}$ | $76.54_{\pm4.06}$ | $\mathbf{71.45}_{\pm6.01}$ | $\mathbf{69.99}_{\pm0.62}$ | $\mathbf{67.18}_{\pm0.51}$ | $64.73_{\pm0.97}$ | $\mathbf{63.04}_{\pm1.35}$ | $61.16_{\pm1.84}$ | $59.04_{\pm1.05}$ | $57.48_{\pm2.90}$ | $\mathbf{57.12}_{\pm0.21}$ |

We also report the AUC (Area Under the Curve) of the accuracy–cost trade-off curve for each method. This is computed using the `sklearn.metrics.auc` function Buitinck et al. [2013]. AUC provides a single scalar summary of performance that captures how well a model balances accuracy and computational cost. A higher AUC indicates a more favorable overall trade-off across budgets, making it a robust evaluation metric for comparing routing strategies.

Table 7: Average AUC over 10 trials across datasets and embedding models. Higher is better. Abbreviations: RB = RouterBench, SP = SPROUT.

| Method | RB-BERT | RB-LLaMa | SP-BERT | SP-LLaMa |
|---|---|---|---|---|
| Full-Feedback | 0.1839 | 0.1719 | 0.1727 | 0.1655 |
| Baseline | 0.0890 | 0.0134 | 0.0255 | 0.0148 |
| R&C | 0.1539 | 0.1108 | 0.1316 | 0.1105 |
| Causal-CARROT (kNN) | 0.1441 | 0.1433 | 0.1642 | 0.1618 |
| Causal-CARROT (EmbedNet) | 0.1376 | 0.0842 | 0.1148 | 0.0768 |
| CF-Regression | 0.1412 | 0.1119 | 0.1352 | 0.1233 |
| RM-Classification | 0.1665 | 0.1389 | 0.1477 | 0.1436 |
| RM-Softmax | **0.2286** | **0.2213** | **0.3320** | **0.2444** |
| RM-Interval | 0.2285 | 0.2196 | 0.3320 | 0.2464 |

## C Contribution of Each Component

The results highlight the impact of causal bias correction, end-to-end training, and regret-focused objectives as follows:

**Causal Inference (Bias Correction):** Our Baseline corresponds to CARROT without any causal correction - that is, a decoupled predictor trained directly on observational data without accounting for treatment bias. It consistently performs the worst across all settings. In contrast, incorporating causal techniques for bias correction yields substantial gains: both R&C and Causal-CARROT, which integrate causal adjustments into routing, achieve +10–15% routing accuracy at comparable cost, demonstrating that accounting for selection bias is critical for effective routing. These results validate that bias-aware routing significantly improves utility over naive predictors trained on biased data.

**End-to-End Learning vs. Two-Stage:** Among bias-corrected approaches, our end-to-end regret-minimizing methods (RM) consistently outperform the two-stage methods (CF-Regression, Causal-CARROT, R&C). The performance gap (+1–3% routing accuracy at comparable cost) demonstrates the benefit of an integrated end-to-end approach: by directly optimizing the decision-quality objective (regret) rather than optimizing intermediate predictions, our method achieves superior and more stable results.

**Regret Minimization Objective:** Even compared to other causal learners, our specific training objective provides an edge. For instance, RM-Softmax (differentiable surrogate) slightly outperforms RM-Classification (upper-bound surrogate) in most cases, with lower variance (+0.5–1% routing accuracy at comparable cost). This highlights the advantage of our softmax-weighted regret objective and its alignment with the true decision loss. Moreover, both RM methods outperform Causal-CARROT and CF-Regression, underscoring that minimizing expected regret is more effective than surrogate approaches that focus only on accuracy/cost prediction.

## D Experimental Details

All experiments were implemented in Python 3.8.12 Van Rossum and Drake [2009], using PyTorch 2.4.1+cu121 Paszke et al. [2019] and Scikit-learn Buitinck et al. [2013]. Experiments were conducted on an internal compute cluster equipped with an Intel(R) Xeon(R) Platinum 8260 CPU @ 2.40GHz, 512 GB of RAM, and two NVIDIA V100 GPUs with 16 GB memory each.

**Prompt Encoding & Augmentation** To encode input queries into vector representations $x$, we employ a two-stage embedding process. First, we enrich each prompt with contextual metadata by prepending a natural language prefix that identifies the source dataset. Specifically, for a prompt p originating from dataset D (e.g., openhermes/teknium), we construct the following context-augmented input: *"The following prompt comes from the dataset D. The prompt is: p"*. This step provides the embedding model with useful dataset-level context, which is particularly beneficial in multi-domain routing scenarios. The template is flexible and can be extended to include additional metadata if desired.

**Datasets.** **RouterBench** [Hu et al., 2024] is a standardized benchmark for LLM routing, comprising 35,712 prompt-response pairs collected from 11 LLMs. The prompts span eight different evaluation benchmarks covering reasoning, factual knowledge, dialogue, mathematics, and code generation. Each prompt is annotated with model accuracy and execution cost, enabling response-based decision-making. To maintain consistency in evaluation, we adopt the same split strategy for RouterBench, applied deterministically at the prompt level to ensure reproducibility. **SPROUT** [Somerstep et al., 2025] is a more recent and larger benchmark for cost-aware routing, consisting of 44,241 prompts and responses from 13 state-of-the-art language models. The prompts are drawn from six diverse benchmarks, including GPQA [Rein et al., 2024], MuSR [Sprague et al., 2023], MMLU-Pro [Wang et al., 2024], MATH [Hendrycks et al., 2021], OpenHermes [Teknium, 2023], and RAGBench [Friel et al., 2024]. SPROUT includes a predefined train/validation/test split, using 80% of the data for training and splitting the remaining 20% equally between validation and test sets.

**Neural Router Models.** All neural models used in our experiments share the same architecture for fairness and comparability. We use a 2-layer feedforward neural network with GELU activation and 200 hidden units per layer. Models are trained using the Adam optimizer with a learning rate of $10^{-4}$, batch size of 128, and a maximum of 10,000 epochs. Early stopping is applied with a patience of 100 epochs based on validation regret. The temperature parameter for the softmax-based regret objective is set to 100, and to 1000 for the interval model to allow smoother gradients across budget intervals.

**Doubly Robust Estimation.** For the outcome model $\hat{r}_t(x)$, we use the same neural architecture described above, trained separately for each treatment $t$. For the propensity model $\hat{p}(t|x)$, we use an XGBoost classifier with the following hyperparameters: maximum depth = [1,2,3,5], number of trees = [10,20,50,100]. The estimated DR scores are clipped to the [5th, 95th] percentile to reduce the impact of extreme propensity weights and improve training stability.

**Embedding Generation.** We generate sentence-level embeddings using the `bert-base-uncased` (768-dim) and `meta-llama/Llama-3.2-1B` (2048-dim) models. Embeddings are extracted via mean pooling over the final hidden states and are precomputed in batches using GPU acceleration. These embeddings are fixed during training of all downstream routing models.

**RM-Interval Network.** The joint model used for budget interpolation is implemented using a small feedforward network that takes as input the concatenation of outputs from $f_{\lambda_j}$, $f_{\lambda_{j+1}}$, and a linear embedding of $\lambda$. The architecture mirrors the router described above and is fine-tuned using the regret objective over interval-specific training data. The proposed architecture is presented in Figure 1.

**Inference latency and computational efficiency** Latency is critical in real-time applications. Our method is designed to minimize this by using lightweight routing networks (2-layer MLP with 200 neurons per hidden layer) and precomputed light-weight embeddings (Llama-3.2-1B and BERT). For instance, with Llama-3.2-1B embeddings and an MLP-based router on the RoutherBench dataset, the end-to-end routing latency is under 2.5 ms on a single A100 GPU for a batch size of almost 25,000. To contextualize this, recent benchmarks show that on 8× A100s, LLaMA-2 or LLaMA-3 70B can generate a 100-token paragraph in 1.5 to 2.5 seconds under realistic conditions using optimized inference stacks. Even when using fewer GPUs or smaller models, generation latency typically remains one to two orders of magnitude higher than our routing time. In many cases (e.g., longer outputs or cold starts), the difference can be significantly larger. Thus, the routing overhead is negligible relative to the cost of LLM generation and does not meaningfully impact user experience. Additionally, the interval-conditioned architecture reduces deployment complexity by avoiding the need to train or store separate models for different user preferences.

