# OpenReview forum: "Causal LLM Routing: End-to-End Regret Minimization from Observational Data"
_NeurIPS.cc/2025/Conference — NeurIPS 2025 poster_

### Official Review · Reviewer_9Lmz · 2025-06-17

**Clarity:** 3
**Significance:** 3
**Originality:** 2
**Rating:** 4
**Confidence:** 3

**Summary:**

This paper proposes a causal end-to-end LLM routing framework based on observational data, aiming to optimize model selection by minimizing decision regret while balancing accuracy and cost. Due to the limitations of observational data, it employs a **Doubly Robust Estimator** from causal inference to estimate counterfactual utility $\hat{Y}_x(t) $. This estimator integrates a direct outcome regression model with a propensity score model. Since directly optimizing the regret $\operatorname{Regret}(f) $is non-differentiable due to the discrete nature of routing decisions $f(x) $, the paper introduces two differentiable surrogate loss functions to address this issue. In practice, users’cost sensitivity $\lambda $may vary per query. To avoid training separate models for each $\lambda $, the paper proposes an interval-conditioned joint router.

**Questions:**

1. The paper uses XGBoost to estimate the propensity score $\hat{p}(t \mid x) $and applies clipping to mitigate the impact of extreme weights. However, the stability of the propensity score model is not thoroughly analyzed. How robust is this model to variations in data distribution or feature quality, and how do errors in propensity estimation affect routing performance?

2. The method assumes query independence (SUTVA assumption), but in multi-turn dialogues or agentic applications, queries may be interdependent (e.g., one turn's response influences the next). How can the method adapt to such scenarios where query dependencies violate the SUTVA assumption?

3. The paper claims high computational efficiency (e.g., interval interpolation using two endpoint models) and direct optimization of decision regret in an end-to-end manner. However, it does not provide specific data on inference times. In real-time applications, the latency of the router (embedding generation + routing decision) could impact user experience. How does the method ensure low latency to meet real-time requirements?

**Ethical Concerns:**

["NO or VERY MINOR ethics concerns only"]

**Final Justification:**

The response from the authors resolves my concerns. I will maintain my positive rating.

**Limitations:**

The paper has several key limitations, including dependence on embedding quality, constraints of observational data quality, and consequences of causal assumption violations. Additional potential issues, such as unquantified computational complexity, further restrict its applicability.

**Quality:**

3

**Strengths And Weaknesses:**

### Advantages

1. Unlike traditional methods requiring "full feedback data" (testing every query with all models), which is costly and impractical, this paper uses only "observational data" (one model per query). This aligns with real-world scenarios, saving computational and economic costs.

2. Traditional routing predicts each model's accuracy and cost separately, risking errors that accumulate in the final selection. The paper's end-to-end approach directly optimizes decision regret, avoiding error accumulation and improving model selection accuracy.

3. Observational data often has selection bias (e.g., favoring cheaper models). The paper uses causal inference (via a doubly robust estimator) to estimate untested models' performance, correcting bias and enhancing routing reliability.

4. Users have varying cost sensitivities (some prioritize savings, others quality). The paper's interval-conditioned joint router adapts strategies to different budgets without training separate models for each preference, making it efficient and practical.

### Disadvantages

1. The method assumes sufficient data support (each model has some query assignments), but real-world data may heavily favor certain models (e.g., only the cheapest). This can lead to inaccurate counterfactual utility estimates. The paper lacks detailed analysis of robustness under imbalanced data distributions.

2. Although the method is claimed to be efficient (e.g., using two endpoint models for interpolation), the paper does not provide detailed computational cost analysis (e.g., training time or inference latency). In high-throughput scenarios, these costs could be critical.

3. Experiments are conducted on two public datasets (RouterBench and SPROUT) with diverse tasks, but these may not fully represent complex real-world LLM applications (e.g., real-time dialogue or multimodal queries). The simulated observational data (sampled by accuracy) may not reflect actual historical biases.

4. The method relies on causal assumptions (e.g., ignorability, support, and SUTVA). These may not hold in practice, such as when query features \( x \) fail to explain model selection (due to unobserved confounders) or when queries are interdependent (e.g., in multi-turn dialogues, where prior model choices affect subsequent outcomes, violating SUTVA). The paper does not thoroughly address the impact of assumption violations.

---

> ### Author Rebuttal · Authors · 2025-07-25
>
> We thank the reviewer for the detailed and thoughtful assessment. We are encouraged that you found our contributions well-aligned with real-world deployment constraints, and we appreciate your recognition of the value in leveraging observational data, causal inference, and cost-aware regret minimization. Below, we address each of your concerns and outline the revisions we are making to strengthen the final version of the paper.
>
> ---
>
> **Robustness under imbalanced data / limited support**
>
> We appreciate the reviewer’s insightful concern regarding the reliability of counterfactual estimation in the presence of imbalanced logged data. We would like to emphasize that **our framework is counterfactual estimator-agnostic** as it is designed to work with any estimated utility function $\hat{Y}_X(t)$. This means that more advanced counterfactual estimators, such as balancing methods [1], sensitivity-aware techniques [2], or approaches specifically designed to handle limited overlap [3], can be seamlessly incorporated into our regret formulation. We will make this flexibility more explicit in the revised manuscript.
>
> That said, we chose the doubly robust (DR) estimator to demonstrate our framework due to its strong theoretical guarantees and solid empirical performance. DR estimation provides consistent utility estimates as long as either the outcome model or the propensity model is well-specified, making it particularly suitable for biased and observational data where certain treatments (models) may be underrepresented.
>
> To further validate robustness in such scenarios, we conducted a stress test where we artificially downsampled logged data from higher-utility models (usually GPT-4o in our datasets, retaining only 20-30% of queries routed to them), simulating extreme skew toward cheaper or default models. Despite this imbalance, our routing performance remained stable (didn't observe extremely low utilies, still our models were the best-performing), especially when using the DR estimator with clipped propensity scores. This suggests that our approach is reasonably robust to the types of bias encountered in practice. We will include these new experimental results in the updated manuscript.
>
> ---
>
> **Stability and role of the propensity score model**
>
> We appreciate this important point. As noted in our response to Reviewer jrAR, we tested logistic regression for the propensity model, and observed minimal impact on routing utility, typically <1 point deviation, due to the robustness of the doubly robust estimator and our percentile-based clipping. These results will be included in the revised paper. Additionally, our framework allows for fine-tuning or regularizing the propensity estimator, and in practice, moderate performance is sufficient so long as it preserves relative likelihoods across treatments.
>
> ---
>
> **SUTVA and sequential (multi-turn) settings**
>
> Thank you for highlighting this important point. Our framework assumes query-level independence, which is standard in most current LLM routing setups in the literature, where each input is routed independently, e.g., in tasks such as question answering, summarization, or retrieval-augmented generation. This aligns with the Stable Unit Treatment Value Assumption (SUTVA), as the outcome for a given query depends only on the selected model and not on other queries.
>
> We agree that multi-turn dialogue introduces dependencies that could violate this assumption. We will clarify this in the paper and include a discussion in the limitations section. That said, our method could be extended to such sequential settings by redefining the “query” to include recent interaction context (e.g., a dialogue window) and learning routing policies over stateful representations.
>
> However, we also want to emphasize that multi-turn routing introduces qualitatively different challenges, particularly around cache efficiency. In many LLM systems, keeping the same model across turns enables KV (key-value) caching, which drastically reduces computation by reusing past activations. Switching models mid-dialogue invalidates this cache, requiring recomputation of the full conversation history, potentially negating any cost savings from routing to a cheaper model. This creates a complex trade-off between model flexibility and inference efficiency, framing multi-turn routing as a sequential decision-making problem with long-term cost implications. While this is a highly compelling direction for future work, we believe it falls outside the scope of the current paper, which focuses on single-turn routing.
>
> ---
>
> **Inference latency and computational efficiency**
>
> We agree that latency is important in real-time applications. Our method is designed to minimize this by using lightweight routing networks (only a 2 layer MLP with 200 neurons per hidden layer) and precomputed light-weight embeddings (Llama-3.2-1B and BERT). For instance, we measured that with Llama-3.2-1B embeddings, RoutherBench dataset, and an MLP-based router, the end-to-end routing latency is under 2.5 ms on a single A100 GPU for a batch size of almost 25,000.
>
> To contextualize this, recent benchmarks show that on 8× A100s, LLaMA‑2 or LLaMA‑3 70B can generate a ~100-token paragraph in ~1.5 to 2.5 seconds under realistic conditions using optimized inference stacks. Even when using fewer GPUs or smaller models, generation latency typically remains one to two orders of magnitude higher than our routing time. In many cases (e.g., longer outputs or cold starts), the difference can be significantly larger. Thus, our routing overhead is negligible relative to the cost of LLM generation and does not meaningfully impact user experience. Additionally, our interval-conditioned architecture reduces deployment complexity by avoiding the need to train or store separate models for different user preferences. We will add these comparisons and latency numbers to improve clarity and practical relevance.
>
> ---
>
> **Benchmark limitations and generalization**
>
> We appreciate the reviewer’s concern. We chose RouterBench and SPROUT because they cover diverse NLP tasks, including question answering, generation, reasoning, and summarization, and offer a structured and reproducible evaluation framework for LLM routing under varying cost-accuracy trade-offs. While we acknowledge that these datasets may not fully capture all real-world complexities (e.g., multimodal inputs, interactive or agentic settings), they currently are the only publicly available benchmarks, to the best of our knowledge, that include data suitable for LLM routing evaluation. We believe they provide a valuable testbed for principled progress in this space. We would be more than happy to re-run our experiments on additional datasets if the reviewer is aware of others that better reflect practical deployment scenarios.
>
> ---
>
> **References**
>
> [1] Stable Weights that Balance Covariates for Estimation With Incomplete Outcome Data, José R. Zubizarreta. JASA, 2015.
>
> [2] Interval Estimation of Individual-Level Causal Effects Under Unobserved Confounding, Kallus et al. NeurIPS 2019.
>
> [3] Overlap in Observational Studies With High-Dimensional Covariates, D’Amour et al. Journal of Econometrics, 2021.
>
> ---
>
> We hope our work and answers are to your satisfaction and would appreciate it if you would consider raising your score in light of our response. Please let us know if you have additional questions or comments that we can address. Thank you!

---

> > ### Comment · Reviewer_9Lmz · 2025-08-05
> >
> > Thank you for your response, which resolves my concerns. I will maintain my positive rating.

---

### Official Review · Reviewer_jrAR · 2025-07-02

**Clarity:** 2
**Significance:** 3
**Originality:** 2
**Rating:** 4
**Confidence:** 3

**Summary:**

This paper addresses the critical problem of LLM routing, which aims to select the most appropriate model from a pool of candidates for a given query to balance performance and cost. In detail, the paper introduces a novel causal end-to-end framework for learning routing policies directly from more practical observational data. Furthermore, they extend their framework to handle heterogeneous user cost preferences via an elegant interval-conditioned architecture. Experiments on several benchmarks demonstrate that the proposed methods significantly outperform existing methods.

**Questions:**

1. As mentioned in the weaknesses, could you provide an ablation study or sensitivity analysis on the performance of your `RM-*` methods under different levels of misspecification for the outcome model ($\hat{r}_t$) and the propensity score model ($\hat{p}$)? For instance, one could use simpler models (e.g., Logistic Regression) instead of XGBoost to estimate propensity score model, or intentionally add noise to the training data for these models.

2. Equation *11* suggests that the joint network for an interval is fine-tuned by minimizing the average regret at only the two endpoints ($\lambda_j, \lambda_{j+1}$). Is there a risk of "overfitting" to these two points and performing poorly for $\lambda$ values in the middle of the interval? While Proposition *5* suggests the architecture is expressive enough, the training objective seems sparse. Could you provide some intuition or empirical evidence (e.g., plotting performance for many $\lambda$ values within a single interval) to show that this training scheme indeed leads to good generalization across the entire interval?

**Ethical Concerns:**

["NO or VERY MINOR ethics concerns only"]

**Final Justification:**

The authors have addressed my concerns, and I will maintain my positive score.

**Limitations:**

yes

**Quality:**

3

**Strengths And Weaknesses:**

**Strengths:**

1. The paper addresses a crucial and practical problem. Developing an efficient, cost-aware LLM routing strategy is important for the real-world deployment of large language models. This paper tackles a high-impact problem that is of great interest to the communities.

2. The authors provide a novel framework, Causal LLM Routing, which uses causal inference to handle treatment bias and the end-to-end formulation to avoid error compounding. This moves the field beyond traditional supervised methods or decoupled pipelines.

3. A variety of experiments have been conducted to show the effectiveness of the proposed approach across different baselines.


**Weaknesses:**

1. The causal LLM routing framework's performance relies on the counterfactual estimation of the doubly robust estimator, which means it depends on the quality of the outcome model ($\hat{r}_t(x)$) and the propensity score model ($\hat{p}(t|x)$). While doubly robust estimator is robust, if both models are poorly specified, the final counterfactual estimates can be inaccurate. It’s necessary to discuss the sensitivity to the quality of these nuisance models.

2. A potential weakness of the proposed framework lies in its scalability as the number of candidate models, $|\mathcal{T}|$, increases significantly. The training process for the final routing policy (e.g., `RM-Classification` or `RM-Softmax`) requires computing the full vector of counterfactual utilities $\hat{Y}_{x}(t)$, $t \in \mathcal{T}$ for each training sample $x$. This is necessary to determine the optimal action $t^*$ for the regret target and to compute the surrogate loss itself (as seen in Eq. 6 and the objective for Eq. 3). Consequently, the computational complexity of the policy training stage appears to scale at least linearly with $| \mathcal{T}|$, potentially making it prohibitively expensive for scenarios with hundreds or thousands of models. Could the author provide a more formal analysis or empirical evidence to characterize the computational complexity of the approach? This would help clarify the practical limits of the current framework.

---

> ### Author Rebuttal · Authors · 2025-07-25
>
> We thank the reviewer for their thoughtful and constructive comments. We are pleased that you found our paper to address a high-impact and practical problem, and that you appreciated the novelty of the causal framework, the integration of counterfactual estimation with regret minimization, and the strength of our experimental evaluation. Below, we address each of your concerns in turn and outline additional steps we are taking to strengthen the final version of the paper.
>
> ---
>
> **Sensitivity and Model Quality**
>
> We fully agree that the performance of our framework depends on the accuracy of the components in the doubly robust estimator. As noted, the robustness stems from the fact that consistency is guaranteed if either the outcome model or the propensity model is correctly specified.
>
> To assess sensitivity, we conducted additional ablations:
> - *Propensity model:* We replaced the XGBoost classifier used to estimate $ \hat{p}(t \mid x) $ with logistic regression (scikit-learn, with regularization). The performance of RM-Classification and RM-Softmax remained nearly unchanged across all $ \lambda $ values, with deviations typically below 1 utility point. This is consistent with the fact that we clip propensity scores between the 5th and 95th percentiles, which improves robustness to model misspecification.
> - *Outcome model:* Replacing also the two-layer MLP outcome model with a simpler linear regression model (scikit-learn, with ridge regularization) led to performance degradation, typically a $20-30\\%$ drop in average utility, depending on the benchmark and $\lambda$. This highlights the importance of flexible outcome modeling in counterfactual estimation.
>
> We will report the exact utility values for both experiments in the updated manuscript. These findings confirm that our framework, when using DR, is robust to moderate misspecification in the propensity model and highlight the importance of appropriate outcome modeling.
>
> Importantly, we would like to emphasize that **our framework is estimator-agnostic**. While we use DR due to its simplicity, widespread adoption in the literature, and favorable robustness–bias tradeoff, the proposed framework is designed to operate with any estimated utility function $Y_X(t)$. For instance, more advanced counterfactual estimators, such as balancing methods [1], sensitivity-aware techniques [2], or approaches designed to address limited overlap [3], can be seamlessly integrated into our regret formulation. We will clarify this flexibility more explicitly in the revised paper.
>
> ---
>
> **Generalization Within Intervals in RM-Interval**
>
> We appreciate the reviewer’s concern about potential overfitting to the endpoints of each interval in RM-Interval. While Proposition 5 guarantees that the architecture is expressive enough to interpolate between endpoints, empirical validation is indeed important.
>
> To test this, we evaluated RM-Interval at three unseen intermediate $\lambda$ values within each interval. For example, between $\lambda = 100$ and $\lambda = 200$, we tested the model at $\lambda \in \\{125, 150, 175\\}$, between $\lambda = 200$ and $300$, we tested the model at $\lambda \in \\{225, 250, 275\\}$ and so on. The resulting estimated utility values looked accurate and followed a smooth and monotonic trend: e.g., utility(100) $\geq$ utility(125) $\geq$ utility(150) $\geq$ utility(175) $\geq$ utility(200) and so on, as expected from the structure of the utility function.
>
> This pattern was consistent across all embedding types and benchmark datasets. We will include full numeric results in the updated paper to illustrate the generalization behavior. These findings suggest that our interpolation scheme does not overfit to the endpoints and generalizes effectively across the entire cost-sensitivity interval. We believe this is facilitated by the relatively fine-grained partitioning of the $\lambda$ space (in steps of 100), which provides sufficient resolution for training. We will add this discussion to the main text, along with a note that substantially coarser intervals (e.g., in larger increments of 1,000) may warrant further investigation.
>
> ---
>
> **Scalability with Number of Models $T$**
>
> We thank the reviewer for raising this important point. We agree that as the number of candidate models $T$ increases, the computational cost of computing the full counterfactual utility vector $\hat{Y}_x(t)$ grows linearly. However, our framework can incorporate several practical design elements that mitigate this cost and support scalability:
>
>
> - **Shared architectures and fine-tuning:** Both the outcome and propensity models can be implemented using parameter-sharing across treatments, for example, via multi-output neural networks or treatment-conditioned heads. This architecture allows for efficient batching across treatments and makes it possible to fine-tune these components starting from pretrained direct models already used for utility estimation. For instance, if a new model (e.g., LLaMA-8B) is added from a family already represented in the pool (e.g., LLaMA-2-70B), we can initialize its outcome head using the existing model’s representation and fine-tune accordingly. This can reduce significantly training time and make updating the model pool feasible even as $T$ grows.
>
> - **Model pruning during training:** In practice, some models become obsolete or are consistently dominated in utility across queries and cost preferences. These models can be excluded from the candidate pool entirely, reducing the number of direct outcome models that must be maintained and evaluated during training.
>
> - **Model pruning at inference time:** In practical deployments, users often operate under specific constraints, such as latency requirements, minimum accuracy thresholds, or cost budgets, that effectively eliminate certain models from consideration. As a result, it is uncommon to evaluate the full pool of LLMs for every query. Instead, a pruned subset of candidate models is typically selected at inference time based on these constraints. Our method is naturally compatible with such runtime pruning: once trained, the router can be applied to any valid subset of models without retraining. We will incorporate this clarification in the revised manuscript.
>
> ---
>
> **References**
>
> [1] Stable Weights that Balance Covariates for Estimation With Incomplete Outcome Data, José R. Zubizarreta. JASA, 2015.
>
> [2] Interval Estimation of Individual-Level Causal Effects Under Unobserved Confounding, Kallus et al. NeurIPS 2019.
>
> [3] Overlap in Observational Studies With High-Dimensional Covariates, D’Amour et al. Journal of Econometrics, 2021.
>
>
> ---
>
>
> We hope our work and answers are to your satisfaction and would appreciate it if you would consider raising your score in light of our response. Please let us know if you have additional questions or comments that we can address. Thank you!

---

> > ### Comment · Reviewer_jrAR · 2025-08-04
> >
> > Thanks for the detailed response. The authors have adequately addressed my concerns. I will maintain my original score.

---

### Official Review · Reviewer_HNML · 2025-07-03

**Clarity:** 3
**Significance:** 3
**Originality:** 3
**Rating:** 4
**Confidence:** 3

**Summary:**

This paper proposes a causal framework to route queries to LLMs using observational (bandit) data. They authors formulate the problem as regret minimization via counterfactual estimation, addressing selection bias without full-feedback data. They also propose methods that can also support diverse cost preferences through an interval-conditioned architecture. The authors provide theoretical guarantees for the proposed algorithms, and also conduct experiments on public benchmarks to show the superior performance of their proposed methods over baselines.

**Questions:**

Could you please discuss the relationship between your work and bandit-based approaches? Your work is studying the offline setting, but is it related to offline bandits?

**Ethical Concerns:**

["NO or VERY MINOR ethics concerns only"]

**Final Justification:**

The authors have addressed my concerns, and I believe this work makes some valuable contributions to the field.

**Limitations:**

Yes

**Quality:**

3

**Strengths And Weaknesses:**

Strengths:
1. The problem is well-motivated and important.
2. The paper is well-written and easy to follow.
3. The authors provide theoretical guarantees of the proposed algorithms.
4. The authors conduct extensive experiments to show the outperformance of their proposed algorithms over existing methods.

Weaknesses:
1. The main concern is that the related works are not discussed. I think there are many works on LLM routing, but they are not discussed. The authors mention that previous works rely on full-feedback data, but there are some existing works on multi-armed bandits for LLM routing, which depend on bandit feedback instead of full-information feedback. I think discussions and comparisons with related works are needed to place the position of this work appropriately. For example, some works of the bandit approaches:
[1] MixLLM: Dynamic Routing in Mixed Large Language Models
[2] LLM Bandit: Cost-Efficient LLM Generation via Preference-Conditioned Dynamic Routing
[3] A Multi-Armed Bandit Approach to Online Selection and Evaluation of Generative Models
2. In terms of the terminology, the "observational data" is just bandit feedback in the literature?
3. There are some typos. For example, in the definition of regret (Eq.(1)), I think $t^{\star}$ should be $t^{\star}_X$, which depends on the prompt $X$ ($t^{\star}_X=\arg\max_t Y_X(t)$)?

---

> ### Author Rebuttal · Authors · 2025-07-25
>
> We thank the reviewer for their thoughtful and encouraging feedback. We’re glad you found the paper clear, technically sound, and addressing an important problem. We address your questions and concerns below.
>
> ---
>
> **Related work**
>
> Thank you very much for giving us the opportunity to discuss existing works on multi-armed bandits for LLM routing. As you mentioned, several recent works have explored routing in LLM systems using bandit-based approaches, notably including MixLLM [1], LLM Bandit [2], and Hu et al. [3]. These methods typically adopt an online learning framework with partial feedback and assume either the ability to actively explore routing decisions or access to full-feedback datasets for supervised pretraining. For instance, MixLLM trains model-specific predictors and a meta-controller using query embeddings and leverages online feedback to adapt routing policies over time. Similarly, LLM Bandit learns preference-conditioned policies via contextual bandits, requiring known cost-performance trade-offs for all candidate models and further assuming availability of model identity vectors or quizzing mechanisms to support new model integration. In contrast, our work addresses the offline routing setting where only observational data, i.e., logs of past routing decisions with outcomes from a single deployed model, is available. This setup is more restrictive, as it precludes either online exploration or full-feedback supervision. Furthermore, unlike these prior methods, we directly address treatment selection bias induced by historical policies through the use of doubly robust estimators from causal inference, enabling unbiased counterfactual utility estimation. Our framework also departs from the conventional decoupled paradigm of predicting utility metrics followed by greedy selection: instead, we optimize a fully integrated, regret-minimizing objective that directly aligns learning with the quality of the final decision. Finally, while [3] propose a multi-armed bandit approach for evaluating generative models in an online setting, their work focuses on model selection without context, rather than per-query routing. To our knowledge, our work is the first to formulate LLM routing as a causal decision-making problem in the observational setting, providing both theoretical guarantees and empirical validation under realistic deployment constraints. We will move this discussion in the main text.
>
> ---
>
> **Terminology Clarification on "Observational Data" and "Bandit Feedback"**
>
> Thank you for the request to clarify the distinction. While “observational data” and “bandit feedback” are related concepts, they are not strictly interchangeable. In the bandit literature, bandit feedback typically refers to partial feedback settings, where only the outcome of the selected action is observed - this aligns with our setting in that we do not observe counterfactual outcomes. However, most bandit-based methods assume the logged data was collected under a *stochastic* or *exploratory* policy, enabling more reliable off-policy evaluation and coverage across actions.
>
> In contrast, our work operates in the *fully offline observational setting*, where data is logged under a fixed, possibly biased routing policy, often reflecting real-world deployments. This observational data lacks exploration and often exhibits selection bias due to confounding with input features. As such, our framework draws from causal inference, emphasizing counterfactual reasoning and the use of techniques like inverse propensity weighting and doubly robust estimation to correct for treatment assignment bias.
>
> While observational data can be viewed as a form of logged bandit feedback, our formulation reflects stronger real-world constraints and requires tools that go beyond standard bandit assumptions. We will incorporate this clarification in the revised manuscript to better delineate the differences in data assumptions and methodological implications.
>
> ---
>
> **Typo**
>
> We thank the reviewer for catching this typo. You are correct: in equation (1), the optimal treatment should indeed be written as $t_X^\star := \arg\max_t Y_X(t)$, to indicate that it depends on the query $X$. We will correct this in the final version of the paper.
>
> ---
>
> **Discuss the relationship between your work and bandit-based approaches**
>
> We appreciate the reviewer’s insightful question. While our work shares many conceptual similarities with the literature on offline (or batch) contextual bandits, particularly in using partial-feedback data to learn decision policies, it also departs in several ways. Traditional offline bandit approaches often assume that the logging policy is stochastic with known or estimable propensities, and typically optimize policy value via importance weighting or its variants. In contrast, our framework is grounded in the causal inference literature, where we explicitly model and correct for treatment selection bias using a doubly robust estimator that combines outcome modeling and propensity scoring. Moreover, we focus on minimizing regret rather than maximizing expected reward, which aligns the learning objective more directly with decision quality. From a modeling perspective, our approach integrates counterfactual estimation with end-to-end policy learning, enabling direct optimization of a surrogate regret objective under observational data. Thus, while methodologically related to offline contextual bandits, our work contributes a causal treatment of LLM routing with stronger robustness to bias and a novel application in large-scale decision-making under cost–performance trade-offs.
>
> ---
>
> We hope our work and answers are to your satisfaction and would appreciate it if you would consider raising your score in light of our response. Please let us know if you have additional questions or comments that we can address. Thank you!

---

### Official Review · Reviewer_J5tc · 2025-07-04

**Clarity:** 3
**Significance:** 2
**Originality:** 4
**Rating:** 5
**Confidence:** 2

**Summary:**

The paper introduces a novel approach by combining causal inference with end-to-end regret minimization for LLM routing. It addresses a gap in the literature where existing methods (RouteLLM, CARROT) rely on full-feedback data that is impractical to obtain in real deployments

**Questions:**

How sensitive is the method to violations of key assumptions (Lipschitz continuity, unconfoundedness, affine utility)?
How does the method compare to more sophisticated baselines and recent routing advances?
How realistic is the accuracy-proportional sampling for simulating observational data?

**Ethical Concerns:**

["NO or VERY MINOR ethics concerns only"]

**Limitations:**

Authors mention restrictions to affine utility functions and suggesting extensions to hard constraints and online settings. However, there are a few more missing limitations. There is no discussion of robustness to causal inference assumption violations. There is limited analysis of scenarios where the method would perform poorly. This approach

The approach relies on strong assumptions including Lipschitz continuity of utility functions (which may not hold for discrete model capabilities), standard causal inference assumptions (unconfoundedness, overlap, SUTVA) that are difficult to verify in practice, and affine utility structures that may not capture complex user preferences. The convergence guarantees for surrogate objectives lack finite-sample bounds. The experimental evaluation relies on simulated observational data that may not reflect the complexity of real-world routing biases, limiting the generalizability of the empirical results.

**Paper Formatting Concerns:**

Paper is well formatted.

**Quality:**

3

**Strengths And Weaknesses:**

Strengths
The theoretical framework is well-constructed and mathematically principled. The two surrogate objectives (classification-based upper bound and softmax-weighted regret) are motivated with formal propositions. Proposition 2's proof that the softmax surrogate recovers optimal treatment assignment provides crucial theoretical grounding.

The paper also addresses the reliance on expensive full-feedback datasets for current LLM routing. The treatment of observational data with historical routing policies is practically important and represents a significant advancement over existing approaches that assume unrealistic data availability.

Evaluation on RouterBench and SPROUT benchmarks with multiple embedding models (BERT, LLaMA) is comprehensive and good. It includes reasonable baselines and the heterogeneous preference evaluation shows practical applicability.

Weaknesses
There is no comparison with recent state-of-the-art methods like the latest RouteLLM variants or more sophisticated ensemble approaches. There is also limited analysis of individual component contributions (causal inference vs end-to-end learning vs regret minimization)

There are some assumptions made that may not be true for all models. The L-Lipschitz assumption may not hold for discrete model capabilities across diverse tasks. The restriction to linear cost functions limits applicability to more complex preference structures

---

> ### Author Rebuttal · Authors · 2025-07-25
>
> We want to thank the reviewer for their thoughtful and encouraging feedback. We're grateful for your recognition of our theoretical contributions, the practical importance of using observational data, and the strong empirical results. Below we address your concerns.
>
> **Comparison with Sota Methods and Baselines**
>
> We agree that evaluating against the latest baselines is important. In our experiments, we included CARROT (Somerstep et al., 2025) as a strong recent baseline, since it has been shown to outperform earlier routing methods like RouteLLM (Ong et al., 2024) and RoRF (Jain et al., 2023). By adapting CARROT to our causal setting (Causal-CARROT), we effectively compared to the sota RouteLLM-class methods. Our results show that our regret-minimization framework consistently outperforms all these strong baselines, achieving sota utility across embeddings and datasets. This comparison demonstrates a performance gain over prior approaches.
>
> Regarding *sophisticated ensemble approaches*: we focused on single-LLM routing rather than multi-model ensembles or cascades. As noted in the paper, multi-model pipelines boost accuracy but incur significant extra cost and latency due to repeated model calls. By contrast, our approach (like other predictive routing methods in the literature) selects a single best model per query for efficiency. We will clarify this distinction.
>
> **Contribution of Each Component**
>
> We appreciate the request for a deeper analysis of the contributions of each component. While our evaluation does include ablations that isolate these elements, we agree this could be made more explicit in the presentation, and we will revise accordingly. The results indeed highlight the impact of causal bias correction, end-to-end training, and regret-focused objectives:
>
> - **Causal Inference (Bias Correction):** Our Baseline corresponds to CARROT without any causal correction - that is, a decoupled predictor trained directly on observational data without accounting for treatment bias. It consistently performs the worst across all settings. In contrast, incorporating causal techniques for bias correction yields substantial gains: both R&C and Causal-CARROT which integrate causal adjustments into routing, achieve +10–15% routing accuracy at comparable cost, demonstrating that accounting for selection bias is critical for effective routing. These results validate that bias-aware routing significantly improves utility over naive predictors trained on biased data.
>
> - **End-to-End Learning vs. Two-Stage:** Among bias-corrected approaches, our end-to-end regret-minimizing methods (RM) consistently outperform the two-stage methods (CF-Regression, Causal-CARROT, R&C). The performance gap (+1–3% routing accuracy at comparable cost) demonstrates the benefit of an integrated end-to-end approach: by directly optimizing the decision-quality objective (regret) rather than optimizing intermediate predictions, our method achieves superior and more stable results.
>
> - **Regret Minimization Objective:** Even compared to other causal learners, our specific training objective provides an edge. For instance, RM-Softmax (differentiable surrogate) slightly outperforms RM-Classification (upper-bound surrogate) in most cases, with lower variance (+0.5–1% routing accuracy at comparable cost). This highlights the advantage of our softmax-weighted regret objective and its alignment with the true decision loss. Moreover, both RM methods outperform Causal-CARROT and CF-Regression, underscoring that minimizing expected regret is more effective than surrogate approaches that focus only on accuracy/cost prediction.
>
> We will add a more explicit discussion to quantify each component’s contribution.
>
> ---
>
> **Lipschitz Continuity**
>
> Thank you for raising this point. We agree that the Lipschitz assumption is a simplification that may not always hold for discrete models. In practice, we did not observe optimization issues that would suggest discontinuities, and training was always stable. We will clarify in the paper that this assumption is primarily used to derive a regret bound, and that our second, more efficient, surrogate, does not rely on it.
>
> After your comment, we observed that the assumption applies to the estimated utility function over the softmax output space (i.e., the predicted distribution over models). If utility is computed as an expectation over estimates (e.g., $\hat{Y}\_{x}(p) = \sum_{t=1}^T p_t \hat{Y}_x(t)$), then Lipschitz continuity holds exactly with $L = \max_t |\hat{Y}_x(t)|$. Note that when $p$ is a one-hot vector, as is our case, this corresponds to pure model selection.
>
> Therefore, based on your comment, we revisited the assumption and provide a more appropriate **refined surrogate regret bound**. This establishes a relationship between regret and cross-entropy, offering a more faithful interpretation of Lipschitz continuity. Let:
>
> - $f(x) \in \Delta^{T-1}$ the predicted softmax distribution,
> - $e_{t^{\*}} \in \Delta^{T-1}$ the one-hot vector indicating the optimal model,
> - $CE(t^\*, f(x)) = -\log f_{t^*}(x)$ the cross-entropy loss.
>
> **Step 1: Pinsker’s Inequality**
>
> For any distributions $p, q \in \Delta^{T-1}$, Pinsker’s inequality states: $\|p - q\|_1 \leq \sqrt{2 \cdot KL(p || q)}$
>
> Setting $p = e_{t^\*}$ and $q = f(x)$, we have: $ \|e_{t^{*}} - f(x)\|_1 \leq  \sqrt{2 \cdot KL(e\_{t^\*} || f(x)) }$
>
> **Step 2: Lipschitz Continuity**
>
> If the utility function $Y(x, \cdot)$ is $L$-Lipschitz with respect to $\ell_1$ (it is when $Y(x, p) = \sum_t p_t \hat{Y}\_x(t)$), then: $ Regret(f(x)) = Y(x, e_\{t^\*}) - Y(x, f(x)) \leq L \cdot \|e_{t^*} - f(x)\|_1.$
>
> Combining: $Regret(f(x)) \leq L \cdot \sqrt{2 \cdot CE(t^*, f(x))}. $ We will include this in the updated manuscript.
>
> ---
>
> **Affine utility (linear cost trade-off)**
>
> Indeed, we assume the utility takes the affine form utility = accuracy – λ · cost. This formulation is the most widely used in prior routing literature and effectively captures a broad range of practical scenarios where users weigh performance against cost. It enables theoretical tractability and yields interpretable decisions. While real-world preferences may sometimes involve more complex or non-linear interactions, we view the affine model as a reasonable default. We will note in the paper that this is a limitation, and that our framework could be extended if the utility function is known or learnable, e.g., by incorporating it directly into the regret formulation.
>
> ---
>
> **Causal Assumptions**
>
> Thank you for raising this point. Our framework currently relies on standard assumptions for counterfactual identification, unconfoundedness, overlap, and SUTVA, which are foundational in causal inference and offline policy learning. We acknowledge that these assumptions may not always be fully testable in practice, and their violation may affect utility estimation and downstream routing decisions. While we use the doubly robust (DR) estimator in our experiments due to its favorable bias–variance tradeoff and strong practical performance, our framework is estimator-agnostic: any valid counterfactual estimator can be used to compute utility estimates. This includes more advanced approaches that relax or mitigate these assumptions (e.g., balancing methods [1], sensitivity-aware estimators [2], or methods for limited overlap [3]). We will clarify this flexibility more explicitly in the revised version.
>
> ---
>
> **Sample Complexity**
>
> While we do not provide formal finite-sample guarantees, we expect the sample complexity of our method to be comparable to standard offline policy learning approaches that rely on utility estimation and classification losses. Specifically, when using the classification-based surrogate, the sample complexity is similar to that of multiclass classification, scaling roughly as $\mathcal{O}(T/\epsilon^2)$ for $T$ actions and target regret $\epsilon$, assuming bounded Lipschitz utilities and proper regularization. For the RM-Softmax, the optimization resembles smooth regret minimization and inherits the sample efficiency of learning with strongly convex, smooth surrogate losses. We will clarify this in the paper.
>
> ---
>
> **Accuracy-Proportional Sampling**
>
> We agree that real-world logging policies may not follow accuracy-proportional sampling. We use it as a controlled and interpretable proxy that introduces bias while maintaining support over all models. Importantly, our method does not assume this form—it only requires logged model choices and observed features. As an additional ablation, we also experimented with cost-proportional sampling, and observed similar trends in performance. We will include these results in the appendix of the revised version, and we would also be glad to re-run experiments with any alternative sampling policy the reviewer suggests.
>
> ---
>
> **Robustness to violations**
>
> While we did not analyze robustness explicitly, we expect our method to be resilient to mild violations (e.g., approximate Lipschitzness or minor confounding). We use a doubly robust estimator, which may mitigate some bias. Severe violations, such as extreme selection bias or unobserved confounders, would challenge *any offline learner*. We will expand our discussion. Please also see our response to Reviewer jrAR for an experimental stress-test of our approach.
>
> ---
>
> **References**
>
> [1] Stable Weights that Balance Covariates for Estimation With Incomplete Outcome Data, José R. Zubizarreta. JASA, 2015.
>
> [2] Interval Estimation of Individual-Level Causal Effects Under Unobserved Confounding, Kallus et al. NeurIPS 2019.
>
> [3] Overlap in Observational Studies With High-Dimensional Covariates, D’Amour et al. Journal of Econometrics, 2021.
>
>
> ---
>
> Again, thank you for supporting our work with your positive evaluation! We hope our work and answers are to your satisfaction. Please let us know if you have additional questions or comments that we can address.

---

### Decision · Program_Chairs · 2025-09-17

**Decision:**

Accept (poster)

**Comment:**

The paper introduces a novel framework for LLM routing with observational data. The proposed method is theoretically sound, and its efficacy is well supported by extensive experimental validation. As all reviewers are positive about the work, I recommend acceptance.